# A Trainable Spectral-Spatial Sparse Coding Model for Hyperspectral Image Restoration

**Théo Bodrito**[*]**, Alexandre Zouaoui**[*]**, Jocelyn Chanussot, and Julien Mairal**
Inria, Univ. Grenoble Alpes, CNRS, Grenoble INP, LJK, 38000 Grenoble, France
`firstname.lastname@inria.fr`

## Abstract

Hyperspectral imaging offers new perspectives for diverse applications, ranging from the monitoring of the environment using airborne or satellite remote sensing, precision farming, food safety, planetary exploration, or astrophysics. Unfortunately, the spectral diversity of information comes at the expense of various sources of degradation, and the lack of accurate ground-truth "clean" hyperspectral signals acquired on the spot makes restoration tasks challenging. In particular, training deep neural networks for restoration is difficult, in contrast to traditional RGB imaging problems where deep models tend to shine. In this paper, we advocate instead for a hybrid approach based on sparse coding principles that retains the interpretability of classical techniques encoding domain knowledge with handcrafted image priors, while allowing to train model parameters end-to-end without massive amounts of data. We show on various denoising benchmarks that our method is computationally efficient and significantly outperforms the state of the art. [1]

## 1 Introduction

Hyperspectral imaging (HSI) enables measurements of the electromagnetic spectrum of a scene on multiple bands (typically about a hundred or more), which offers many perspectives over traditional color RGB imaging. For instance, the high-dimensional information present in a single pixel is sometimes sufficient to identify the signature of a particular material, which is of course infeasible in the RGB domain. Not surprisingly, hyperspectral imaging is then of utmost importance and has a huge number of scientific and technological applications such as remote sensing [8, 22, 46], quality evaluation of food products [16, 19, 35], medical imaging [2, 18, 37], agriculture and forestry [1, 36, 40], microscopy imaging in biology [25, 56], or exoplanet detection in astronomy [24].

Information contained in hyperspectral signals is much richer than in RGB images, but the price to pay is the need to deal with complex degradations that may arise from multiple sources, including sparse noise with specific patterns (stripes), in addition to photon and thermal noise [29, 51]. As a consequence, HSI denoising is a crucial pre-processing step to enhance the image quality before using data in downstream tasks such as semantic segmentation or spectral unmixing [30]. A second issue is the lack of large-scale collection of ground-truth high-quality signals and the large diversity of sensor types, which makes it particularly challenging to train machine learning models for restoration such as convolutional neural networks. To deal with the scarcity of ground-truth data, most successful approaches typically encode strong prior knowledge about data within the model architecture, which may be low-rank representations of input patches [17, 23, 53, 60, 70], sparse coding [13, 21, 23], or image self-similarities [39, 49, 71], which have proven to be very powerful in the RGB domain [9].

---

[*]Equal contribution

[1]Code is available at `https://github.com/inria-thoth/T3SC`.

In this paper, we propose a fully interpretable machine learning model for hyperspectral images that may be seen as a hybrid approach between deep learning techniques, where parameters can be learned end to end with supervised data, and classical methods that essentially rely on image priors. Since designing an appropriate image prior by hand is very hard, our goal is to benefit from deep learning principles (here, differentiable programming [6]) while encoding domain knowledge and physical rules about hyperspectral data directly into the model architecture, which we believe is a key to develop robust approaches that do not require massive amounts of training data.

More precisely, we introduce a novel trainable spectral-spatial sparse coding model with two layers, which performs the following operations: (i) The first layer decomposes the spectrum measured at each pixel as a sparse linear combination of a few elements from a learned dictionary, thus performing a form of linear spectral unmixing per pixel, where dictionary elements can be seen as basis elements for spectral responses of materials present in the scene. (ii) The second layer builds upon the output of the first one, which is represented as a two-dimensional feature map, and sparsely encodes patches on a dictionary in order to take into account spatial relationships between pixels within small receptive fields. To further reduce the number of parameters to learn and leverage classical prior knowledge about spectral signals [60], we also assume that the dictionary elements admit a low-rank structure—that is, dictionary elements are near separable in the space and spectrum domains, as detailed later. Even though dictionary learning has been originally introduced for unsupervised learning [42, 47], we adopt an unrolled optimization procedure inspired by the LISTA algorithm [26], which has been very successful in imaging problems for training sparse coding models from supervised data [33, 34, 55, 63].

Our motivation for adopting a two-layer model is to provide a shared architecture for different HSI sensors, which often involve a different number of bands with different spectral responses. Our solution consists of learning sensor-specific dictionaries for the first layer, while the dictionary of second layer is shared across modalities. This allows training simultaneously on several HSI signals, the first layer mapping input data to a common space, before processing data by the second layer.

We experimentally evaluate our HSI model on standard denoising benchmarks, showing a significant improvement over the state of the art (including deep learning models and more traditional baselines), while being computationally very efficient at test time. Perhaps more important than pure quantitative results, we believe that our work also draws interesting conclusions for machine learning. First, by encoding prior knowledge within the model architecture directly, we obtain models achieving excellent results with a relatively small number of parameters to learn, a conclusion also shared by [33, 34] for RGB imaging; nevertheless, the effect is stronger in our work due to the scarcity of training data for HSI denoising and the difficulty to train deep learning models for this task. Second, we also show that interpretable architectures are useful: our model architecture can adapt to different noise levels per band and modify the encoding function at test time in a principled manner, making it well suited for solving blind denoising problems that are crucial for processing hyperspectral signals.

## 2 Related Work on Hyperspectral Image Denoising

**Learning-free and low-rank approaches.** Classical image denoising methods such as BM3D [12] may be applied independently to each spectral band of HSI signals, but such an approach fails to capture relations between channels; Not surprisingly, multi-band techniques such as BM4D [39] have been shown to perform better for HSI, and other variants were subsequently proposed such as GLF [71]. Tensor-based methods such as LLRT [11] are able to exploit the underlying low-rank structure of HSI signals [17, 53, 67] and have shown particularly effective when combined with a non-local image prior as in NGMeet [27]. Finally, other approaches adapt traditional image processing priors such as total variation [65, 59], or wavelet sparsity [48, 52] but they tend to perform worse than GLF, LLRT, or NGMeet, see [31] for a survey on denoising techniques for HSI.

**Sparse coding models.** Dictionary learning [47] is an unsupervised learning technique consisting of representing a signal as a linear combination of a few elements from a learned dictionary, which has shown to be very effective for various image restoration tasks [15, 43]. Several approaches have then combined dictionary learning and low-rank regularization. For instance, 3D patches are represented as tensors in [49] and are encoded by using spatial-spectral dictionaries [57]. In [70], 2D patches are extracted from the band-vectorized representation of the 3D HSI data and sparsely encoded on a dictionary, while encouraging low-rank representations with a trace norm penalty on

the reconstructed image. The low-rank constraint can also be enforced by designing the dictionary as the result of the matrix multiplication between spatial and spectral dictionaries learned by principal component analysis as in [21]. However, these methods typically compute sparse representations with an iterative optimization procedure, which may be computationally demanding at test time.

**Deep learning.** Like BM3D above, convolutional neural networks for grayscale image denoising (*e.g.*, DnCNN [68]) may also be applied to each spectral band, which is of course suboptimal. Because deep neural networks have been highly successful for RGB images with often low computational inference cost, there have been many attempts to design deep neural networks dedicated to HSI denoising. For instance, to account for the large number of hyperspectral bands, several approaches based on convolutional neural networks are operating on sliding windows in the spectral domain, [38, 54, 66], which allows training models on signals with different number of spectral bands, but the sliding window significantly increases the inference time. More precisely, attention layers are used in [54], while more traditional CNNs are used in [38], possibly with residual connections [66]. Recently, an approach based on recurrent architecture was proposed in [62] to process signals with an arbitrary number of bands, achieving impressive results for various denoising tasks.

**Hybrid approaches.** SMDS-Net [63] adopts a hybrid approach between sparse coding and deep learning models by adapting the RGB image restoration method of [34] to HSI images. The resulting pipeline however lacks interpretability: SMDS-Net first denoises the input image with non-local means [9], then performs subspace projection in the spectral domain using HySime[7], before sparsely encoding 3D patches (cubes) with a trainable version of Tensor-based ISTA [50]. Although this method reduces considerably the number of parameters in comparison to vanilla deep learning models, the spectral sliding windows approach lacks interpretability since the same denoising procedure is applied across different bands, which may not suffer from the same level of noise. In contrast, we propose a much simpler sparse coding model, which is physically consistent with the nature of hyperspectral signals, by introducing a novel differentiable low-rank sparse coding layer.

## 3  A Trainable Spectral-Spatial Sparse Coding Model (T3SC)

In this section, we introduce our trainable spectral-spatial sparse coding model dedicated to hyperspectral imaging, and start by presenting some preliminaries on sparse coding.

### 3.1  Background on Sparse Coding

**Image denoising with dictionary learning.** A classical approach introduced by Elad and Aharon [15] for image denoising consists in considering the set of small overlapping image patches (*e.g.*, $8 \times 8$ pixels) from a noisy image, and compute a sparse approximation of these patches onto a learned dictionary. The clean estimates for each patch are then recombined to produce the full image.

Formally, let us consider a noisy image $\mathbf{y}$ in $\mathbb{R}^{c \times h \times w}$ with $c$ channels and two spatial dimensions. We denote by $\mathbf{y}_1, \mathbf{y}_2, \cdots \mathbf{y}_n$ the $n$ overlapping patches from $\mathbf{y}$ of size $c \times s \times s$, which we represent as vectors in $\mathbb{R}^m$ with $m = cs^2$. Assuming that a dictionary $\mathbf{D} = [\mathbf{d}_1, \cdots, \mathbf{d}_p]$ in $\mathbb{R}^{m \times p}$ is given—we will discuss later how to obtain a "good" dictionary— each patch $\mathbf{y}_i$ is processed by computing a sparse approximation:

$$\min_{\boldsymbol{\alpha}_i \in \mathbb{R}^p} \frac{1}{2} \|\mathbf{y}_i - \mathbf{D}\boldsymbol{\alpha}_i\|^2 + \lambda \|\boldsymbol{\alpha}_i\|_1, \tag{1}$$

where $\| \cdot \|_1$ is the $l_1$-norm, which is known to induce sparsity in the problem solution [42], and $\boldsymbol{\alpha}_i$ is the sparse code representing the patch $\mathbf{y}_i$, while $\lambda$ controls the amount of regularization. Note that the $\ell_0$-penalty, which counts the number of non-zero elements, could also be used, leading to a combinatorial problem whose solution is typically approximated by a greedy algorithm. After solving the $n$ problems (1), each patch $\mathbf{y}_i$ admits a "clean" estimate $\mathbf{D}\boldsymbol{\alpha}_i$. Because each pixel belongs to several patches, the full denoised image $\hat{\mathbf{x}}$ is obtained by averaging these estimates.

Finding a good dictionary can be achieved in various manners. In classical dictionary learning algorithms, $\mathbf{D}$ is optimized such that the sum of the loss functions (1) is as small as possible, see [42] for a review. Adapting the dictionary with supervision is also possible [41], as discussed next.

**Differentiable programming for sparse coding.** The proximal gradient descent method called ISTA [20] is a classical algorithm for solving the Lasso problem in Eq. (1), which consists of the following iterations

$$\boldsymbol{\alpha}_i^{(t+1)} = S_\lambda \left[ \boldsymbol{\alpha}_i^{(t)} + \eta \mathbf{D}^\top \left( \mathbf{y}_i - \mathbf{D}\boldsymbol{\alpha}_i^{(t)} \right) \right], \tag{2}$$

where $\eta > 0$ is a step-size and $S_\lambda[u] = \text{sign}(u) \max(|u| - \lambda, 0)$ is the soft-thresholding operator, which is applied pointwise to each entry of an input vector.

By noting that the above iteration can be seen as a sequence of affine transformations interleaved with pointwise non-linearities $S_\lambda$, it is then tempting to interpret $T$ iterations (2) as a multilayer feed-forward neural network with shared weights between the $T$ layers. Following such an insight, Gregor and LeCun have proposed the LISTA algorithm [26], where the parameters are learned such that the sequence approximates well the solution of the sparse coding problem (1).

Interestingly, the LISTA algorithm can also be used to train dictionaries for supervised learning tasks. This is the approach chosen in [34, 55] for image restoration, which considers the following iterations:

$$\boldsymbol{\alpha}_i^{(t+1)} = S_\lambda \left[ \boldsymbol{\alpha}_i^{(t)} + \mathbf{C}^\top \left( \mathbf{y}_i - \mathbf{D}\boldsymbol{\alpha}_i^{(t)} \right) \right], \tag{3}$$

which differs from (2) with the presence of a matrix $\mathbf{C}$ of the same size as $\mathbf{D}$. Even if the choice $\mathbf{C} = \eta \mathbf{D}$ (which recovers ISTA) is perfectly reasonable, using a different dictionary $\mathbf{C}$ has empirically shown to provide improvements in results quality [34], probably due to faster convergence of the LISTA iterations. Then, given some fixed parameters $\mathbf{C}, \mathbf{D}$, a clean estimate $\mathbf{W}\boldsymbol{\alpha}_i^{(T)}$ for each patch $\mathbf{y}_i$ is obtained by using a dictionary $\mathbf{W}$, where $T$ is the number of LISTA steps. The reason for allowing a different dictionary $\mathbf{W}$ than $\mathbf{D}$ is to correct the potential bias due to $\ell_1$-minimization.

Finally, the denoised image $\hat{\mathbf{x}}$ is reconstructed by averaging the patch estimates:

$$\hat{\mathbf{x}} = \frac{1}{m} \sum_{i=1}^{n} \mathbf{R}_i \mathbf{W} \boldsymbol{\alpha}_i^{(T)}, \tag{4}$$

where $\mathbf{R}_i$ is the linear operator that places the patch $\hat{\mathbf{x}}_i$ at position $i$ in the image, and we assume—by neglecting border effects for simplicity—that each pixel admits the same number $m$ of estimates.

In contrast to classical restoration techniques based on dictionary learning, the LISTA point of view enables us to learn the model parameters $\mathbf{C}, \mathbf{D}, \mathbf{W}$ in a supervised fashion. Given a training set of pairs of noisy/clean images, we remark that the estimate $\hat{\mathbf{x}}$ is obtained from a noisy image $\mathbf{y}$ by a sequence of operations that are differentiable almost everywhere, as typical neural networks with rectified linear unit activation functions. A typical loss, which we optimize by stochastic gradient descent, is then

$$\min_{\mathbf{C}, \mathbf{D}, \mathbf{W}, \lambda} \mathbb{E}_{\mathbf{x}, \mathbf{y}} \left[ \| \hat{\mathbf{x}}(\mathbf{y}) - \mathbf{x} \|^2 \right],$$

where $(\mathbf{x}, \mathbf{y})$ is a pair of clean/noisy images drawn from some training distribution from which we can sample, and $\hat{\mathbf{x}}(\mathbf{y})$ is the clean estimate obtained from (4), given the noisy image $\mathbf{y}$.

## 3.2 A Trainable Low-Rank Sparse Coding Layer

We are now in shape to introduce a trainable layer encoding both sparsity and low-rank principles.

**Spatial-Spectral Representation.** As shown in [10, 21], HSI patches can be well reconstructed by using only a few basis elements obtained by principal component analysis. The authors further decompose these into a Cartesian product of separate spectral and spatial dictionaries. In this paper, we adopt a slightly different approach, where we consider a single dictionary $\mathbf{D} = [\mathbf{d}_1, \ldots, \mathbf{d}_p]$ in $\mathbb{R}^{m \times p}$ as in the previous section with $m = cs^2$, but each element may be seen as a matrix of size $c \times s^2$ with low-rank structure. More precisely, we enforce the following representation

$$\forall j \in 1, \ldots, p, \ \mathbf{d}_j = \text{vec}\left( \mathbf{U}_j \times \mathbf{V}_j \right), \tag{5}$$

where $\mathbf{U}_j$ is in $\mathbb{R}^{s^2 \times r}$, $\mathbf{V}_j$ is in $\mathbb{R}^{r \times c}$, $r$ is the desired rank of the dictionary elements, and $\text{vec}(.)$ is the operator than flattens a matrix to a vector. The hyperparameter $r$ is typically small with $r = 1, 2$ or $3$. When $r = 1$, the dictionary elements are said to be separable in the spectral and spatial domains, which we found to be a too stringent condition to achieve good reconstruction in practice.

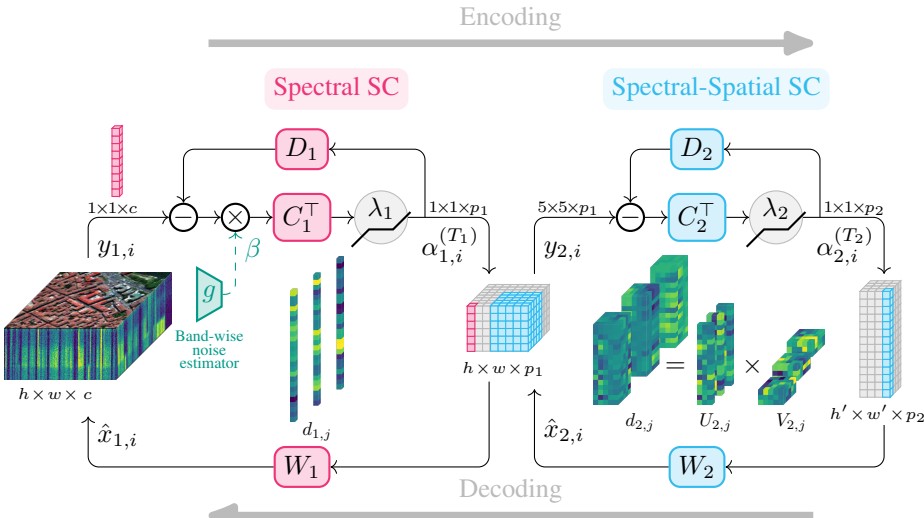

Figure 1: Architecture of T3SC : we propose a two-layer sparse coding model which is end-to-end trainable. The first layer performs a sensor-specific spectral decomposition, while the second layer encodes both spectral and spatial information.

The low-rank assumption allows us to build model with a reduced number of parameters, while encoding natural assumption about the data directly in the model architecture. Indeed, whereas a classical full-rank dictionary $\mathbf{D}$ admits $cs^2p$ parameters, the decomposition (5) yields dictionaries with $(s^2 + c)rp$ parameters only. Matrices $\mathbf{C}$ and $\mathbf{W}$ are parametrized in a similar manner.

**Convolutional variant and implementation tricks.** Whereas traditional sparse coding reconstructs local signals (patches) independently according to the iterations (3), another variant called convolutional sparse coding (CSC) represents the whole image by a sparse linear combination of dictionary elements placed at every possible location in the image [55]. From a mathematical point of view, the reconstruction loss for computing the codes $\boldsymbol{\alpha}_i$ given an input image $\mathbf{y}$ becomes

$$\min_{\{\boldsymbol{\alpha}_i \in \mathbb{R}^p\}_{i=1,\dots,n}} \frac{1}{2} \left\| \mathbf{y} - \frac{1}{m} \sum_{i=1}^{n} \mathbf{R}_i \mathbf{D} \boldsymbol{\alpha}_i \right\|^2 + \lambda \sum_{i=1}^{n} \|\boldsymbol{\alpha}_i\|_1. \tag{6}$$

An iterative approach for computing these codes can be obtained by a simple modification of (3) consisting of replacing the quantity $\mathbf{D}\boldsymbol{\alpha}_i^{(t)}$ by the $i$-th patch of the reconstructed image $\frac{1}{m}\sum_{i=1}^{n} \mathbf{R}_i \mathbf{D}\boldsymbol{\alpha}_i^{(t)}$. All of these operations can be efficiently implemented in standard deep learning frameworks, since the corresponding operations corresponds to a transposed convolution with $\mathbf{D}$, followed by convolution with $\mathbf{C}$, see [55] for more details. In this paper, we experimented with the CSC variant (6) and SC one (1), both with low-rank dictionaries, which were previously described. We observed that CSC was providing slightly better results and was thus adopted in our experiments. Following [34], another implementation trick we use is to consider a different $\lambda$ parameter per dictionary element, which slightly increases the number of parameters, while allowing to learn with a weighted $\ell_1$-norm in (6).

### 3.3 The Two-Layer Sparse Coding Model with Sensor-Specific Layer

One of the main challenge in hyperspectral imaging is to train a model that can generalize to several types of sensors, which typically admit different number of spectral bands. Whereas learning a model that is tuned to a specific sensor is perfectly acceptable in many contexts, it is often useful to learn a model that is able to generalize across different types of HSI signals. To alleviate this issue, several strategies have been adopted such as (i) projecting signals onto a linear subspace of fixed dimension, with no guarantee that representations within this subspace can be comparable between different signals, or (ii) processing input data using a sliding window across the spectral domain.

In this paper, we address this issue by learning a two-layer model, presented in Figure 1, where the first layer is tuned to a specific sensor, whereas the second layer could be generic. Note that the

second layer carries most of the model parameters (about $20\times$ more than in the first layer in our experiments). Formally, let us denote by $\boldsymbol{\alpha}$ in $\mathbb{R}^{p \times h \times w}$ the sparse encoding of an input tensor $\mathbf{y}$ in $\mathbb{R}^{c \times h \times w}$ as previously described. A sparse coding layer $\Phi$ naturally yields an encoder and a decoder such that:

$$\Phi^{enc} : \mathbf{y} \mapsto \boldsymbol{\alpha}, \quad \text{and} \quad \Phi^{dec} : \boldsymbol{\alpha} \mapsto \frac{1}{n} \sum_{i=1}^{n} \mathbf{R}_i \mathbf{W} \boldsymbol{\alpha}_i. \quad (7)$$

Given a noisy image $\mathbf{y}$, the denoising procedure described in the previous section with one layer can be written as

$$\hat{\mathbf{x}}(\mathbf{y}) = \Phi^{dec} \circ \Phi^{enc}(\mathbf{y}).$$

Then, a straightforward multilayer extension of the procedure may consist of stacking several sparse coding layers $\Phi_1, \ldots, \Phi_L$ together to form a multilayer sparse coding denoising model:

$$\hat{\mathbf{x}}(\mathbf{y}) = \Phi_1^{dec} \circ \cdots \circ \Phi_L^{dec} \circ \Phi_L^{enc} \circ \cdots \circ \Phi_1^{enc}(\mathbf{y}).$$

The model we propose is composed of two layers, as shown in Figure 1. The first layer encodes spectrally the input HSI image, meaning that it operates on $1 \times 1$ patches, whereas the second layer encodes both spectrally and spatially the output of the first layer.

### 3.4 Noise Adaptive Sparse Coding

An advantage of using a model based on a sparse coding objective (1) is to give the ability to encode domain knowledge within the model architecture. For instance, the Lasso problem (1) seen from a maximization a posteriori estimator implicitly assumes that the noise is i.i.d. If the noise variance is different on each spectral band, a natural modification to the model is to introduce weights and use a weighted-$\ell_2$ data fitting term (which could be applied as well to the CSC model of (6)):

$$\min_{\boldsymbol{\alpha}_i \in \mathbb{R}^p} \frac{1}{2} \sum_{j=1}^{c} \beta_j \|\mathbf{M}_j(\mathbf{y}_i - \mathbf{D}\boldsymbol{\alpha}_i)\|^2 + \lambda\|\boldsymbol{\alpha}_i\|_1, \quad (8)$$

where $\mathbf{M}_j$ is a linear operator that extracts band $j$ from a given HSI signal. From a probabilistic point of view, if $\sigma_j^2$ denotes the variance of the noise for band $j$, we may choose the corresponding weight $\beta_j$ to be proportional to $1/\sigma_j^2$. Yet, estimating accurately $\sigma_j^2$ is not always easy, and we have found it more effective to simply learn a parametric function $\beta_j = g(\mathbf{M}_j\mathbf{y})$—here, a very simple CNN with three layers, see supplementary material for details—which is applied independently to each band. It is then easy to modify the LISTA iterations accordingly to take into account these weights, and learn the model parameters jointly with those of the parametric function $g$.

### 3.5 Self-Supervised Learning: Blind-Band Denoising with No Ground Truth Data

Even though acquiring limited ground truth data for a specific sensor is often feasible, it is also interesting to be able to train models with no ground truth at all, *e.g.*, for processing images without physical access to the sensor. In such an unsupervised setting, deep neural networks are typically trained for RGB images by using blind-spot denoising techniques [32], consisting of predicting pixel values given their context. Here, we propose a much simpler approach exploiting the spectral redundancy between channels. More precisely, each time we draw an image for training, we randomly mask one band (ore more), and train the model to reconstruct the missing band from the available ones. Formally, the training objective becomes

$$\min_{\mathbf{C}, \mathbf{D}, \mathbf{W}, \lambda} \mathbb{E}_{\mathbf{x}, \mathbf{y}, S} \left[ \sum_{j \notin S} \|\mathbf{M}_j(\hat{\mathbf{x}}_S(\mathbf{y}) - \mathbf{y})\|^2 \right], \quad (9)$$

where $S$ is the set of bands that are visible for computing the sparse codes $\boldsymbol{\alpha}_i$, leading to a reconstructed image that we denote by $\hat{\mathbf{x}}_S(\mathbf{y})$. Formally, it would mean considering the objective (8), but replacing the sum $\sum_{j=1}^{c}$ by $\sum_{j \in S}$. This is in spirit similar to blind-spot denoising, except that bands are masked instead of pixels, making the resulting implementation much simpler.

Table 1: Simplified comparison between learning-free and learning-based approaches.

|                    | *Data req.* | *training*  | *inference* | *adapt. to new data* | *complex noise* |
|--------------------|-------------|-------------|-------------|----------------------|-----------------|
| **learning-free**  | no req.     | no training | slow        | easy                 | poor            |
| **learning-based** | clean data  | slow        | fast        | complicated          | good perf.      |

## 4    Experiments

We now present various experiments to demonstrate the effectiveness of our approach for HSI denoising, but first, we discuss the difficulty of defining the state of the art in this field. We believe indeed that it is not always easy to compare learning-free from approaches based on supervised learning. These two classes of approaches have very different requirements/characteristics, making one class more relevant than the other one in some scenarios, and less in others. Table 1 summarizes their characteristics, displaying advantages and drawbacks of both approaches.

**Benchmarked models.**    Keeping in mind the previous dichotomy, we choose to compare our method to traditional methods such as bandwise BM3D [12] (implementation based on [44, 45]), BM4D [39], GLF [71], LLRT [11], NGMeet [27]. We also included deep learning models in our benchmark such as HSID-CNN [66], HSI-SDeCNN [38] 3D-ADNet [54], SMDS-Net [63] and QRNN3D [62]. Results of HSID-CNN, HSI-SDeCNN and 3D-ADNet on Washington DC Mall (available in the Appendix) are taken directly from the corresponding papers, as the train/test split is the same. Otherwise, the results were obtained by running the code obtained directly from the authors, except for SMDS-Net, where our implementation turned out to be slightly more effective. Note that the same architecture for our model was used in all our experiments (see Appendix).

**Datasets.**    We evaluate our approach on two datasets with significantly different properties.

- *ICVL* [4] consists of 204 images of size $1392 \times 1300$ with 31 bands. We used 100 images for training and 50 for testing as in [62] but with a different train/test split ensuring that similar images—*e.g.*, picture from the same scene—are not used twice.
- *Washington DC Mall* is perhaps the most widely used dataset[2] for HSI denoising and consists of a high-quality image of size $1280 \times 307$ with 191 bands. Following [54], we split the image into two sub-images of size $600 \times 307$ and $480 \times 307$ for training and one sub-image of size $200 \times 200$ for testing. Even though the test image does not overlap with train images, they nevertheless share common characteristics. Interestingly, the amount of training data is very limited here.

Specific experiments were also conducted with the datasets APEX [28], Pavia[3], Urban[58] and CAVE [64], which appear in the supplementary material.

**Normalization.**    Before denoising, HSI images are normalized to $[0, 1]$. For remote sensing datasets, we pre-compute the 2[nd] and 98[th] percentiles for each band, on the whole the training set. Then, normalization is performed on train and test images by clipping each band between those percentiles before applying bandwise min-max normalization, similar to [5, 38]. For the close-range dataset ICVL, we simply apply global min-max normalization as in [63, 62].

**Noise patterns.**    We evaluate our model against different types of synthetic noise:

- *i.i.d Gaussian noise with known variance* $\sigma^2$, which is the same on all bands.
- *Gaussian noise with unknown band-dependent variance*: We consider Gaussian noise with different standard deviation $\sigma_j$ for each band, which is uniformly drawn in a fixed interval. These standard deviations change from an image to the other and are unknown at test time.
- *Noise with spectrally correlated variance*: We consider Gaussian noise with standard deviation $\sigma_j$ varying continuously across bands, following a Gaussian curve, see details in the appendix.
- *Stripes noise* : similar to [62], we applied additive stripes noise to 33% of bands. In those bands, 10-15% of columns are affected, meaning a value uniformly sampled in the interval $[-0.25, 0.25]$ is added to them. Moreover, all bands are disturbed by Gaussian noise with noise intensity $\sigma = 25$.

---

[2]`https://engineering.purdue.edu/~biehl/MultiSpec/hyperspectral.html`
[3]`http://www.ehu.eus/ccwintco/index.php?title=Hyperspectral_Remote_Sensing_Scenes`

**Metrics.** In order to assess the performances the previous methods, we used five different indexes widely used for HSI restoration, namely (i) Mean Peak Signal-to-Noise Ratio (MPSNR), which is the classical PSNR metric averaged across bands; (ii) Mean Structural Similarity Index Measurement (MSSIM), which is based on the SSIM metric [61]; (iii) Mean Feature Similarity Index Measurement (MFSIM) introduced in [69]; (iv) Mean ERGAS [14], and (v) Mean Spectral Angle Map (MSAM) [3]. We use MPSNR and MSSIM in the main paper and report the other metrics in the appendix.

**Implementation details.** We trained our network by minimizing the MSE between the groundtruth and restored images. For ICVL, we follow the training procedure described in [62]: we first center crop training images to size $1024 \times 1024$, then we extract patches of size $64 \times 64$ at scales 1:1, 1:2, and 1:4, with stride 64, 32 and 32 respectively. The number of extracted patches for ICVL amounts to 52962. For Washington DC Mall, we do not crop training images and the patches are extracted with stride 16, 8 and 8, for a total of 1650 patches. One epoch in Washington DC Mall corresponds to 10 iterations on the training dataset. Basic data augmentation schemes such as $90°$ rotations and vertical/horizontal flipping are performed. Code and additional details about optimization, implementation, computational resources, are provided in the supplementary material. As reported in Table 4, augmenting the number unrolled iterations improves the denoising performances at the expense of inference time. Since the Spectral-Spatial SC layer is the most time-consuming, the number of unrolled iterations chosen for the first and second layers are 12 and 5 respectively.

**Quantitative results on synthetic noise.** We present in Table 2 the results obtained on the ICVL dataset (results on DCMall are presented in the appendix). Our method uses the vanilla model of Section 3.3 for the experiments with constant $\sigma$ or correlated noise. For the blind denoising experiment with band-dependent $\sigma$ or for the stripe noise experiment, we use the variant of Section 3.4, which is designed to deal with unknown noise level per channel. The method "T3SC-SSL" implements the self-supervised learning approach of Section 3.5, which does not rely on ground-truth data.

- Our supervised approach achieves state-of-the-art results (or is close to the best performing baseline) on all settings. GLF performs remarkably well given that this baseline is learning-free.
- Our self-supervised method achieves a relatively good performance under i.i.d. Gaussian noise, but does not perform as well under more complex noise. This is a limitation of the approach which is perhaps expected and overcoming this limitation would require designing a different self-supervised learning scheme; this is an interesting problem, which is beyond the scope of this paper.

A visual result on ICVL is shown in Figure 2 for stripes noise. Inference times are provided in Table 3, showing that our approach is computationally efficient.

**Results on real noise.** We also conducted a denoising experiment on the Urban dataset, reporting a visual result in Figure 3. Deep models were pre-trained on the APEX dataset, which has the same number of channels as Urban (even though the sensors are different), with band-dependent noise with $\sigma \in [0-55]$. Please note that for this experiment we did not use Noise Adaptive Sparse Coding3.4 for T3SC, as it is highly dependent on the type of sensor used for training. We show that learning-based models trained on synthetic noise are able to transfer to real data.

**Comments on the additional results presented in the appendix.** The appendix also contains (i) results on the DCMall dataset including additional baselines mentioned above; (ii) error bars for parts of our experimental results in order to assess their statistical significance; (iii) an experiment when learning simultaneously on several datasets with different types of sensors showing that the second layer can be generic and effective at the same time; (iv) additional visual results; (v) various ablation studies to illustrate the importance of different components of our method.

**Broader Impact**

Our paper addresses the problem of denoising the signal, which is a key pre-processing step before using hyperspectral signals in concrete applications. As such, it is necessarily subject to dual use. For instance, HSI may be used for environmental monitoring, forestry, yield estimation in agriculture, natural disaster management planning, astronomy, archaeology, and medicine. Yet, HSI is also used by the petroleum industry for finding new oil fields, and has obvious military applications for surveillance. We believe the potential benefits of HSI for society are large enough to outweigh the

Table 2: Denoising performance on ICVL with various types of noise patterns. The first four rows correspond to i.i.d. Gaussian noise with fixed $\sigma$ per band. The next three rows corresponds to a noise level that depends on the band, taken uniformly on small interval. This is a blind-noise experiment since at test time, the noise level is unknown. The last two rows correspond to the scenarios with correlated $\sigma$ across bands, and with stripe noise, respectively. See main text for details.

| $\sigma$ | Metrics | Noisy | BM3D | BM4D | GLF | LLRT | NGMeet | SMDS | QRNN3D | T3SC | T3SC-SSL |
|---|---|---|---|---|---|---|---|---|---|---|---|
| 5 | MPSNR | 34.47 | 46.17 | 48.85 | 51.25 | 51.86 | **52.74** | 50.91 | 48.80 | 52.62 | 51.42 |
|   | MSSIM | 0.7618 | 0.9843 | 0.9916 | 0.9949 | 0.9951 | **0.9960** | 0.9944 | 0.9918 | 0.9959 | 0.9952 |
| 25 | MPSNR | 21.44 | 37.86 | 39.89 | 43.16 | 43.43 | 44.74 | 42.83 | 44.20 | **45.38** | 44.73 |
|   | MSSIM | 0.1548 | 0.9269 | 0.9510 | 0.9695 | 0.9746 | 0.9796 | 0.9700 | 0.9782 | **0.9825** | 0.9805 |
| 50 | MPSNR | 16.03 | 34.22 | 34.22 | 39.26 | 39.69 | 41.08 | 39.25 | 41.67 | **42.16** | 41.62 |
|   | MSSIM | 0.0502 | 0.8654 | 0.8654 | 0.9197 | 0.9504 | 0.9602 | 0.9382 | 0.9655 | **0.9677** | 0.9646 |
| 100 | MPSNR | 10.85 | 30.43 | 32.47 | 34.79 | 36.39 | 37.55 | 35.64 | 37.19 | **38.99** | 38.50 |
|   | MSSIM | 0.0144 | 0.7557 | 0.8155 | 0.7982 | 0.9182 | 0.9311 | 0.8815 | 0.9140 | **0.9439** | 0.9394 |
| [0-15] | MPSNR | 33.89 | 45.81 | 45.35 | 50.57 | 48.50 | 41.67 | 48.23 | 52.07 | **53.31** | 51.26 |
|   | MSSIM | 0.6386 | 0.9767 | 0.9735 | 0.9948 | 0.9899 | 0.9078 | 0.9900 | 0.9957 | **0.9967** | 0.9955 |
| [0-55] | MPSNR | 23.36 | 39.06 | 38.43 | 44.22 | 41.13 | 32.94 | 41.76 | 47.13 | **48.64** | 46.82 |
|   | MSSIM | 0.2601 | 0.9231 | 0.9074 | 0.9818 | 0.9580 | 0.7565 | 0.9620 | 0.9884 | **0.9911** | 0.9882 |
| [0-95] | MPSNR | 19.06 | 36.17 | 35.55 | 41.43 | 38.44 | 29.40 | 38.94 | 43.98 | **46.30** | 44.75 |
|   | MSSIM | 0.1614 | 0.8760 | 0.8540 | 0.9674 | 0.9354 | 0.6609 | 0.9357 | 0.9753 | **0.9859** | 0.9822 |
| Corr. | MPSNR | 28.85 | 42.73 | 42.13 | 47.05 | 45.76 | 38.06 | 45.98 | 48.90 | **49.89** | 48.78 |
|   | MSSIM | 0.4740 | 0.9599 | 0.9070 | 0.9881 | 0.9824 | 0.8536 | 0.9835 | 0.9911 | **0.9923** | 0.9911 |
| Strip. | MPSNR | 21.20 | 34.88 | 37.70 | 42.06 | 39.38 | 39.78 | 41.98 | 44.60 | **44.74** | 43.80 |
|   | MSSIM | 0.1508 | 0.8641 | 0.9198 | 0.9628 | 0.9258 | 0.9333 | 0.9655 | **0.9806** | 0.9805 | 0.9773 |

Table 3: Inference time per image on ICVL with $\sigma = 50$; SMDS, QRNN3D and T3SC are using a V100 GPU; BM4D, GLF, LLRT and NGMeet are using an Intel(R) Xeon(R) CPU E5-1630 v4 @ 3.70GHz. Note that unlike GLF, NGMeet, and LRRT, learning-based approaches such as QRNN3D and our approach require a training procedure, which may be conducted offline. The cost of such a training step was about 13.5 hours for our method and 19 hours for QRNN3D on a V100 GPU.

| | BM3D | BM4D | GLF | LLRT | NGMeet | SMDS | QRNN3D | T3SC | T3SC-SSL |
|---|---|---|---|---|---|---|---|---|---|
| Inference time (s) | 1677 | 2382 | 5565 | 24384 | 2686 | 74.3 | **3.6** | 5.8 | 54.2 |

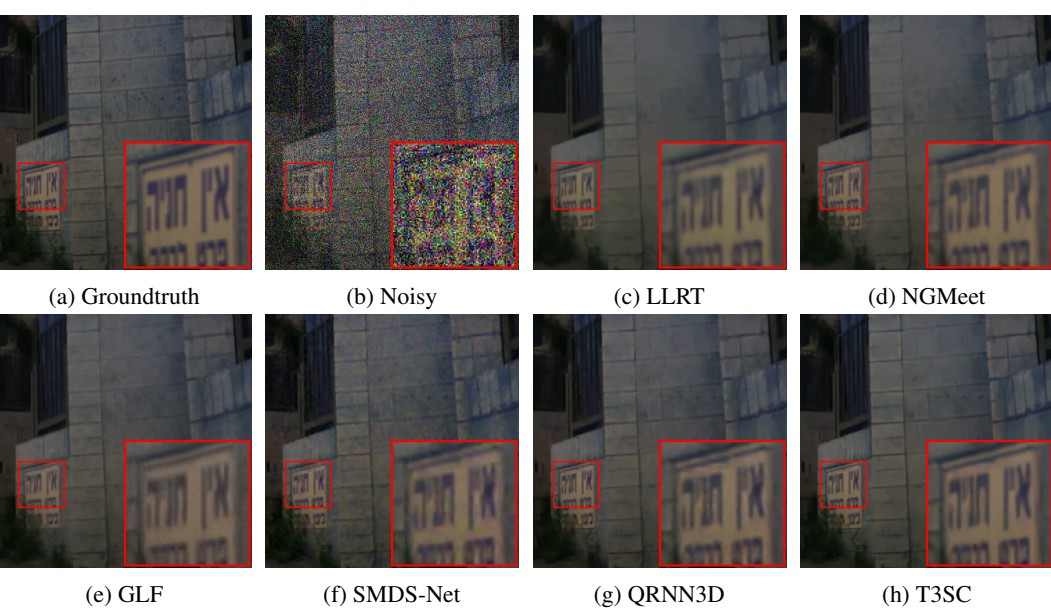

(a) Groundtruth    (b) Noisy    (c) LLRT    (d) NGMeet

(e) GLF    (f) SMDS-Net    (g) QRNN3D    (h) T3SC

Figure 2: Denoising results with Gaussian noise $\sigma = 25$ on ICVL with bands 9, 15, 28.

Table 4: Impact of the number of unrolled iterations per layer on denoising performances and inference time. This experiment was carried out on ICVL with $\sigma = 50$.

| Unrolled iterations per layer | 1 | 2 | 5 | 12 |
|---|---|---|---|---|
| MPSNR | 40.16 | 41.48 | 42.15 | 42.45 |
| Inference time (s) | 0.38 | 1.44 | 5.27 | 14.91 |

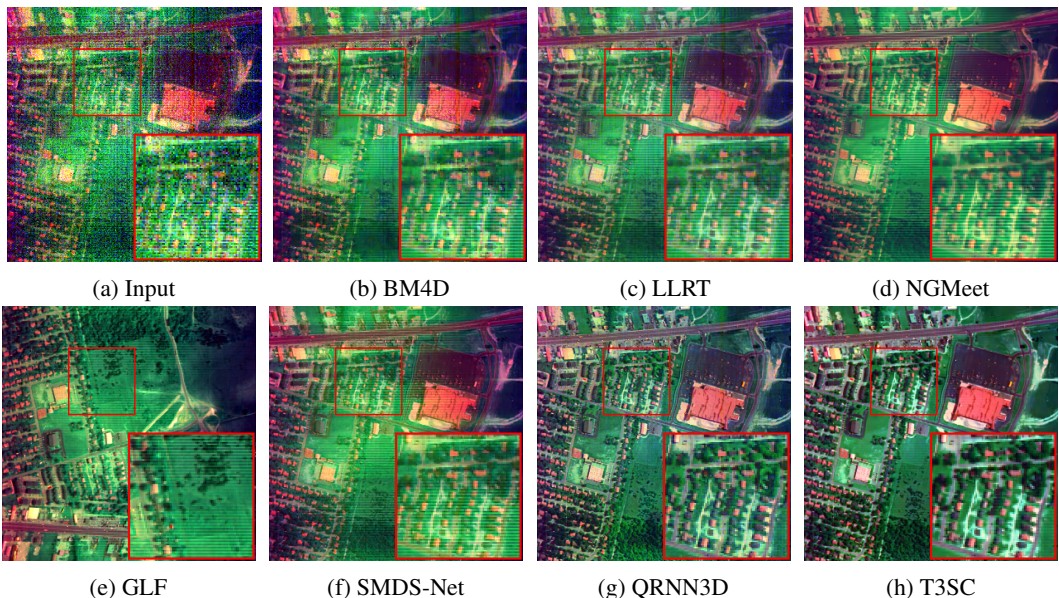

| (a) Input | (b) BM4D | (c) LLRT | (d) NGMeet |

| (e) GLF | (f) SMDS-Net | (g) QRNN3D | (h) T3SC |

Figure 3: Visual result on a real HSI denoising experiment on Urban dataset with bands 1, 108, 208.

potential harm. Nevertheless, we are planning to implement appropriate dissemination strategies to mitigate the risk of misuse for this work (notably with restrictive software licenses), while targeting a gold standard regarding the scientific reproducibility of our results.

**Acknowledgments and Funding**

This project was supported by the ERC grant number 714381 (SOLARIS project) and by ANR 3IA MIAI@Grenoble Alpes (ANR-19-P3IA-0003). This work was granted access to the HPC resources of IDRIS under the allocation 2021-[AD011012382] made by GENCI.

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
