# Supplementary Material
## A Trainable Spectral-Spatial Sparse Coding Model
## for Hyperspectral Image Restoration

## A    Implementation details

In this section, we provide additional implementation details, which are useful to reproduce our experiments (note that the code is also provided).

**Noise with spectrally correlated variance.**    For each band $i \in [\![0, c-1]\!]$, the standard deviation of the Gaussian noise is defined as :

$$\sigma_i = \beta \, exp \left[ -\frac{1}{4\eta^2} \left( \frac{i}{c} - \frac{1}{2} \right)^2 \right]$$

with $\beta = 23.08$ and $\eta = 0.157$.

**Preprocessing.**    A basic centering step is used for each input patch of our model. More precisely, for the first layer, each band of the input hyperspectral image is centered independently prior to patches extraction, and means are added back after decoding. For the second layer, patches are centered independently for each band (and similarly, the means are added back after decoding).

**Code and patch sizes**    The hyperparameters of our model are presented in Table 1.

| Layer | Patches size | Code size | Unrolled iterations | Rank |
|-------|:---:|:---:|:---:|:---:|
| Spectral SC | $1 \times 1$ | 64 | 12 | 1 |
| Spectral-Spatial SC | $5 \times 5$ | 1024 | 5 | 3 |

Table 1: Architecture of our model

Table 10 shows that the combination of both layers is more performant than each layer independently.

**Initialization**    All parameters are initialized with He initialization [1].

**Blocks inference**    In order to apply our model to large images, we split them into blocks of size $256 \times 256$ with an overlap of 6 pixels. Each block is denoised independently. The output image is obtained by aggregating the denoised blocks. Pixels comprised in several blocks are averaged.

**Weights estimator**    For complex noise such as Gaussian noise with band-dependent variance or stripes noise, our model uses a CNN to estimate the weights $\beta_j$ associated with each band. The CNN operates on centered patches of size $56 \times 56$ both during training (random crops) and inference (blocks inference), and its architecture is described in Table 2.

| Layer | Kernel size | Stride | #filters | Output size |
|-------|:---:|:---:|:---:|:---:|
| Inputs | | | | $1 \times 56 \times 56$ |
| Conv2D + ReLU | 5 | 2 | 64 | $64 \times 26 \times 26$ |
| MaxPooling2D | | | | $64 \times 13 \times 13$ |
| Conv2D + ReLU | 3 | 2 | 128 | $128 \times 6 \times 6$ |
| MaxPooling2D | | | | $128 \times 3 \times 3$ |
| Conv2D + Sigmoid | 3 | 1 | 1 | $1 \times 1 \times 1$ |

Table 2: CNN architecture for estimating $\beta_j$

The ablation study presented in Table 11 shows that this extension improves performances substantially for complex noise.

**Optimization** Our models are trained with batch size of 16 for 60 epochs. We use the Adam optimizer, the initial learning rate is $3 \times 10^{-4}$, and is divided by two at epoch 30 and 45.

**Self-Supervised Learning** During the training, $n$ bands randomly selected are masked simultaneously, and reconstructed from the available ones. The MSE loss is applied on the masked bands only. For testing, the $n$ masked bands are evenly distributed along the spectral dimension. All bands are reconstructed after $\lceil c/n \rceil$ iterations, where $c$ denotes the total number of bands and $\lceil \cdot \rceil$ is the ceiling operator. We used $n = 4$ for ICVL and $n = 16$ for Washington DC Mall.

For the SSL setting to be realistic, the noise is added to the clean image before patches are extracted. Otherwise, the model would have indirect access to the groundtruth by seeing the same patch with different noise realizations. As a consequence, the denoising task is much harder on complex noise when data is limited, as shown in Table 3.

# B   Additional Quantitative Results.

**Washington DC Mall dataset.** Results for this dataset are presented in Table 3. Additional baselines are presented in Table 4. The conclusions are similar to those already drawn in the main paper.

Table 3: Denoising performances on Washington DC Mall.

| $\sigma$ | Metrics | Noisy | BM3D | BM4D | GLF | LLRT | NGMeet | SMDS | QRNN3D | T3SC | T3SC-SSL |
|---|---|---|---|---|---|---|---|---|---|---|---|
| 5 | MPSNR | 34.31 | 35.10 | 41.13 | 39.57 | 41.83 | 37.57 | 42.83 | 43.42 | **43.85** | 42.56 |
| | MSSIM | 0.9821 | 0.9875 | 0.9962 | 0.9953 | 0.9968 | 0.9928 | 0.9971 | 0.9973 | **0.9978** | 0.9967 |
| 25 | MPSNR | 20.70 | 24.51 | 31.08 | 35.25 | 34.95 | 35.38 | 35.64 | 35.04 | **36.74** | 35.92 |
| | MSSIM | 0.7688 | 0.8859 | 0.9690 | 0.9883 | 0.9863 | 0.9886 | 0.9889 | 0.9864 | **0.9912** | 0.9894 |
| 50 | MPSNR | 15.25 | 20.80 | 26.69 | 31.77 | 30.94 | 31.88 | 31.76 | 31.72 | **33.12** | 31.96 |
| | MSSIM | 0.5314 | 0.7508 | 0.9220 | 0.9761 | 0.9704 | 0.9759 | 0.9765 | 0.9741 | **0.9819** | 0.9762 |
| 100 | MPSNR | 10.48 | 17.65 | 22.51 | 27.81 | 26.82 | 27.86 | 28.02 | 27.41 | **29.48** | 28.04 |
| | MSSIM | 0.2888 | 0.5427 | 0.8141 | 0.9475 | 0.9322 | 0.9460 | 0.9491 | 0.9375 | **0.9618** | 0.9460 |
| [0-15] | MPSNR | 33.32 | 34.62 | 37.22 | 39.89 | 40.04 | 37.40 | 40.77 | **43.72** | 41.83 | 38.16 |
| | MSSIM | 0.9551 | 0.9746 | 0.9903 | 0.9950 | 0.9951 | 0.9926 | 0.9958 | **0.9971** | 0.9968 | 0.9917 |
| [0-55] | MPSNR | 22.45 | 26.11 | 29.04 | 38.37 | 33.36 | 32.55 | 34.31 | 38.44 | **39.28** | 31.93 |
| | MSSIM | 0.7450 | 0.8683 | 0.9504 | 0.9934 | 0.9811 | 0.9780 | 0.9859 | 0.9925 | **0.9945** | 0.9781 |
| [0-95] | MPSNR | 18.18 | 23.06 | 25.77 | 36.98 | 30.07 | 29.21 | 30.80 | 35.84 | **37.20** | 27.79 |
| | MSSIM | 0.5889 | 0.7688 | 0.9033 | 0.9914 | 0.9643 | 0.9589 | 0.9718 | 0.9877 | **0.9920** | 0.9561 |
| Corr. | MPSNR | 28.48 | 30.50 | 33.69 | 37.96 | 37.77 | 36.56 | 38.54 | 39.84 | **40.79** | 39.61 |
| | MSSIM | 0.9085 | 0.9515 | 0.9637 | 0.9928 | 0.9921 | 0.9911 | 0.9934 | 0.9944 | **0.9960** | 0.9944 |
| Strip. | MPSNR | 20.47 | 24.08 | 29.07 | 35.27 | 34.13 | 34.94 | 35.24 | 35.25 | **36.34** | 34.50 |
| | MSSIM | 0.7621 | 0.8672 | 0.9433 | 0.9877 | 0.9833 | 0.9876 | 0.9876 | 0.9874 | **0.9906** | 0.9853 |

Table 4: Denoising performances on Washington DC Mall with additional baselines.

| $\sigma$ | Metrics | Noisy | BM3D | BM4D | GLF | LLRT | NGMeet | 3D-ADNet | HSID-CNN | HSI-SDeCNN | SMDS-Net | QRNN3D | T3SC |
|---|---|---|---|---|---|---|---|---|---|---|---|---|---|
| 5 | MPSNR | 34.31 | 35.10 | 41.13 | 39.57 | 41.83 | 37.57 | 42.08 | 41.68 | 39.98 | 42.83 | 43.42 | **43.85** |
| | MSSIM | 0.9821 | 0.9875 | 0.9962 | 0.9953 | 0.9968 | 0.9928 | 0.9968 | 0.9966 | 0.9954 | 0.9971 | 0.9973 | **0.9978** |
| 25 | MPSNR | 20.70 | 24.51 | 31.08 | 35.25 | 34.95 | 35.38 | 33.78 | 33.05 | 33.44 | 35.64 | 35.04 | **36.74** |
| | MSSIM | 0.7688 | 0.8859 | 0.9690 | 0.9883 | 0.9863 | 0.9886 | 0.9825 | 0.9813 | 0.9822 | 0.9889 | 0.9864 | **0.9912** |
| 50 | MPSNR | 15.25 | 20.80 | 26.69 | 31.77 | 30.94 | 31.88 | 29.73 | 28.96 | 29.61 | 31.76 | 31.72 | **33.12** |
| | MSSIM | 0.5314 | 0.7508 | 0.9220 | 0.9761 | 0.9704 | 0.9759 | 0.9587 | 0.9536 | 0.9608 | 0.9765 | 0.9741 | **0.9819** |
| 100 | MPSNR | 10.48 | 17.65 | 22.51 | 27.81 | 26.82 | 27.86 | 24.74 | 25.29 | 25.75 | 28.02 | 27.41 | **29.48** |
| | MSSIM | 0.2888 | 0.5427 | 0.8141 | 0.9475 | 0.9322 | 0.9460 | 0.9064 | 0.9014 | 0.9121 | 0.9491 | 0.9375 | **0.9618** |

**Study of statistical significance for the ICVL dataset.** In order to evaluate the statistical significance of our results, we present some results in Table 5 for some of our models and baselines, by running models with five different random seeds. Note that we did not conduct such a study for all results in this paper in order to keep the computational cost of the project reasonable. The conclusions of the paper remain unchanged.

Table 5: Denoising performances on ICVL with multiple seeds

| $\sigma$ | Metrics | Noisy | GLF | NGMeet | SMDS | QRNN3D | T3SC | T3SC-SSL |
|---|---|---|---|---|---|---|---|---|
| 5 | MPSNR | $34.47 \pm 0.01$ | $51.25 \pm 0.01$ | $\mathbf{52.74 \pm 0.01}$ | $50.78 \pm 0.09$ | $49.54 \pm 1.28$ | $\underline{52.62 \pm 0.01}$ | $51.37 \pm 0.03$ |
|  | MSSIM | $0.7619 \pm 0.0001$ | $0.9951 \pm 0.0001$ | $\mathbf{0.9961 \pm 0.0001}$ | $0.9943 \pm 0.0001$ | $0.9924 \pm 0.0021$ | $\underline{0.9960 \pm 0.0001}$ | $0.9952 \pm 0.0001$ |
| 25 | MPSNR | $21.43 \pm 0.01$ | $43.16 \pm 0.01$ | $\underline{44.74 \pm 0.01}$ | $42.63 \pm 0.11$ | $44.20 \pm 0.16$ | $\mathbf{45.37 \pm 0.02}$ | $44.70 \pm 0.02$ |
|  | MSSIM | $0.1548 \pm 0.0002$ | $0.9696 \pm 0.0001$ | $0.9797 \pm 0.0001$ | $0.9687 \pm 0.0009$ | $0.9780 \pm 0.0009$ | $\mathbf{0.9825 \pm 0.0001}$ | $\underline{0.9805 \pm 0.0001}$ |
| 50 | MPSNR | $16.03 \pm 0.01$ | $39.26 \pm 0.01$ | $41.09 \pm 0.01$ | $39.09 \pm 0.08$ | $41.47 \pm 0.14$ | $\mathbf{42.16 \pm 0.01}$ | $\underline{41.62 \pm 0.01}$ |
|  | MSSIM | $0.0503 \pm 0.0001$ | $0.9198 \pm 0.0002$ | $0.9603 \pm 0.0001$ | $0.9359 \pm 0.0012$ | $0.9639 \pm 0.0012$ | $\mathbf{0.9677 \pm 0.0001}$ | $\underline{0.9648 \pm 0.0001}$ |
| 100 | MPSNR | $10.85 \pm 0.01$ | $34.78 \pm 0.01$ | $37.55 \pm 0.01$ | $35.59 \pm 0.04$ | $38.38 \pm 0.60$ | $\mathbf{38.99 \pm 0.01}$ | $\underline{38.51 \pm 0.01}$ |
|  | MSSIM | $0.0144 \pm 0.0001$ | $0.7981 \pm 0.0004$ | $0.9312 \pm 0.0001$ | $0.8781 \pm 0.0017$ | $0.9370 \pm 0.0114$ | $\mathbf{0.9439 \pm 0.0002}$ | $\underline{0.9397 \pm 0.0001}$ |
| [0-15] | MPSNR | $33.94 \pm 0.09$ | $50.68 \pm 0.11$ | $41.57 \pm 0.14$ | $48.00 \pm 0.13$ | $\underline{52.10 \pm 0.12}$ | $\mathbf{53.10 \pm 0.12}$ | $51.21 \pm 0.11$ |
|  | MSSIM | $0.6381 \pm 0.0013$ | $0.9950 \pm 0.0001$ | $0.9065 \pm 0.0022$ | $0.9899 \pm 0.0001$ | $\underline{0.9958 \pm 0.0001}$ | $\mathbf{0.9966 \pm 0.0001}$ | $0.9955 \pm 0.0002$ |
| [0-55] | MPSNR | $23.41 \pm 0.09$ | $44.41 \pm 0.12$ | $32.93 \pm 0.09$ | $41.42 \pm 0.18$ | $\underline{47.26 \pm 0.12}$ | $\mathbf{48.57 \pm 0.28}$ | $46.47 \pm 0.23$ |
|  | MSSIM | $0.2621 \pm 0.0025$ | $0.9820 \pm 0.0004$ | $0.7534 \pm 0.0031$ | $0.9593 \pm 0.0015$ | $\underline{0.9889 \pm 0.0004}$ | $\mathbf{0.9915 \pm 0.0005}$ | $0.9856 \pm 0.0024$ |
| [0-95] | MPSNR | $19.11 \pm 0.09$ | $41.62 \pm 0.11$ | $29.40 \pm 0.12$ | $38.86 \pm 0.06$ | $\underline{44.07 \pm 0.08}$ | $\mathbf{46.24 \pm 0.24}$ | $43.98 \pm 0.46$ |
|  | MSSIM | $0.1644 \pm 0.0031$ | $0.9667 \pm 0.0007$ | $0.6601 \pm 0.0051$ | $0.9352 \pm 0.0004$ | $\underline{0.9758 \pm 0.0003}$ | $\mathbf{0.9863 \pm 0.0005}$ | $0.9735 \pm 0.0049$ |

**CAVE dataset.** We report denoising performances of T3SC on the CAVE Dataset in Table 6 To evaluate T3SC, the dataset was divided in four splits : three were used for training and one for testing. The values reported for T3SC are averaged across all rotations of the test split.

Table 6: Denoising performances on CAVE dataset with Gaussian noise.

| $\sigma$ | Metrics | Noisy | NGMeet | T3SC |
|---|---|---|---|---|
| 5 | MPSNR | 35.05 | 47.96 | **49.16** |
| 25 | MPSNR | 21.99 | 42.44 | **42.77** |
| 50 | MPSNR | 16.37 | 38.89 | **39.7** |
| 100 | MPSNR | 10.96 | 34.99 | **36.48** |

**Joint training across heterogeneous datasets.** In Table 7, we study the problem of training a single model on three different datasets, APEX, DC Mall, and Pavia, involving a different number of channels. As mentioned in the paper, this model involves a common second layer and a spectral dictionary per dataset. These result show that most of the model parameters (which are present in the second layer) can in fact be shared across datasets without significant loss of accuracy when compared to the training of three different models (thus involving three times more parameters).

Table 7: Results for joint training experiment

| Training procedure | Model | Metrics | APEX | DC Mall | Pavia Center |
|---|---|---|---|---|---|
| Independant trainings | QRNN3D | MPSNR | 33.19 | 31.72 | 30.56 |
|  |  | MSSIM | 0.9619 | 0.9741 | 0.9569 |
|  | T3SC | MPSNR | **34.91** | **33.12** | **31.32** |
|  |  | MSSIM | **0.9730** | **0.9819** | **0.9617** |
| Joint training | QRNN3D | MPSNR | 31.95 | 30.97 | 29.12 |
|  |  | MSSIM | 0.9501 | 0.9690 | 0.9428 |
|  | T3SC | MPSNR | **34.74** | **33.08** | **31.30** |
|  |  | MSSIM | **0.9711** | **0.9819** | **0.9616** |

**Additional metrics.** Additional metrics are provided for the ICVL and DCMall datasets, respectively in Tables 8 and 9. The conclusions of the paper are unchanged.

| $\sigma$ | Metrics | Noisy | BM3D | BM4D | GLF | LLRT | NGMeet | SMDS | QRNN3D | T3SC | T3SC-SSL |
|---|---|---|---|---|---|---|---|---|---|---|---|
| | MFSIM | 0.9953 | 0.9978 | 0.9986 | 0.9994 | 0.9995 | **0.9996** | 0.9993 | 0.9987 | **0.9996** | 0.9995 |
| 5 | MERGAS | 6.18 | 1.48 | 1.10 | 0.84 | 0.7740 | **0.69** | 0.87 | 1.14 | 0.70 | 0.83 |
| | MSAM | 0.2460 | 0.0518 | 0.0390 | 0.0267 | 0.0229 | **0.0211** | 0.0307 | 0.0412 | 0.0223 | 0.0286 |
| | MFSIM | 0.9218 | 0.9773 | 0.9829 | 0.9944 | 0.9942 | 0.9954 | 0.9921 | 0.9967 | **0.9970** | 0.9965 |
| 25 | MERGAS | 27.33 | 3.86 | 3.21 | 2.13 | 2.19 | 1.77 | 2.20 | 1.86 | **1.65** | 1.79 |
| | MSAM | 0.5989 | 0.1286 | 0.1005 | 0.0595 | 0.0459 | **0.0384** | 0.0717 | 0.0537 | 0.0406 | 0.0501 |
| | MFSIM | 0.8100 | 0.9488 | 0.9488 | 0.9851 | 0.9851 | 0.9863 | 0.9782 | **0.9928** | 0.9925 | 0.9914 |
| 50 | MERGAS | 51.48 | 5.88 | 5.88 | 3.33 | 3.92 | 2.71 | 3.33 | 2.50 | **2.40** | 2.56 |
| | MSAM | 0.7546 | 0.1964 | 0.1964 | 0.1029 | 0.0682 | **0.0505** | 0.1033 | 0.0571 | 0.0549 | 0.0663 |
| | MFSIM | 0.6471 | 0.8942 | 0.9008 | 0.9679 | 0.9637 | 0.9661 | 0.9456 | **0.9835** | 0.9824 | 0.9805 |
| 100 | MERGAS | 95.97 | 9.11 | 7.96 | 5.59 | 6.22 | 4.08 | 5.04 | 4.20 | **3.46** | 3.66 |
| | MSAM | 0.8619 | 0.2984 | 0.2228 | 0.1847 | 0.0919 | **0.0679** | 0.1441 | 0.1009 | 0.0761 | 0.0889 |
| | MFSIM | 0.9876 | 0.9954 | 0.9963 | 0.9991 | 0.9985 | 0.9965 | 0.9984 | 0.9995 | **0.9996** | 0.9993 |
| [0-15] | MERGAS | 10.11 | 1.91 | 2.07 | 0.98 | 1.17 | 4.53 | 1.20 | 0.79 | **0.69** | 0.91 |
| | MSAM | 0.3412 | 0.0680 | 0.0672 | 0.0328 | 0.0311 | 0.1772 | 0.0408 | 0.0265 | **0.0234** | 0.0322 |
| | MFSIM | 0.9087 | 0.9743 | 0.9768 | 0.9950 | 0.9900 | 0.9755 | 0.9890 | 0.9984 | **0.9985** | 0.9978 |
| [0-55] | MERGAS | 33.34 | 4.17 | 4.73 | 2.07 | 3.02 | 14.69 | 2.50 | 1.39 | **1.20** | 1.52 |
| | MSAM | 0.6478 | 0.1443 | 0.1412 | 0.0687 | 0.0636 | 0.4086 | 0.0784 | 0.0427 | **0.0370** | 0.0502 |
| | MFSIM | 0.8291 | 0.9524 | 0.9560 | 0.9911 | 0.9798 | 0.9536 | 0.9772 | 0.9969 | **0.9972** | 0.9962 |
| [0-95] | MERGAS | 54.92 | 5.83 | 6.73 | 2.86 | 4.64 | 24.82 | 3.46 | 2.17 | **1.58** | 1.92 |
| | MSAM | 0.7720 | 0.2001 | 0.1928 | 0.0992 | 0.0813 | 0.5574 | 0.1042 | 0.0622 | **0.0471** | 0.0615 |
| | MFSIM | 0.9704 | 0.9902 | 0.9923 | 0.9981 | 0.9968 | 0.9919 | 0.9969 | 0.9990 | **0.9991** | 0.9988 |
| Corr. | MERGAS | 14.20 | 2.61 | 3.74 | 1.46 | 1.63 | 6.37 | 1.55 | 1.12 | **1.02** | 1.15 |
| | MSAM | 0.4617 | 0.0934 | 0.1540 | 0.0468 | 0.0416 | 0.2550 | 0.0515 | 0.0316 | **0.0291** | 0.0367 |
| | MFSIM | 0.9068 | 0.9579 | 0.9736 | 0.9926 | 0.9871 | 0.9880 | 0.9900 | **0.9968** | 0.9965 | 0.9956 |
| Strip. | MERGAS | 28.14 | 7.65 | 4.65 | 2.52 | 4.34 | 4.32 | 2.44 | 1.78 | **1.77** | 2.00 |
| | MSAM | 0.6067 | 0.2197 | 0.1442 | 0.0764 | 0.1272 | 0.1298 | 0.0790 | **0.0439** | 0.0534 | 0.0631 |

Table 8: Additional metrics on ICVL

| $\sigma$ | Metrics | Noisy | BM3D | BM4D | GLF | LLRT | NGMeet | SMDS | QRNN3D | T3SC | T3SC-SSL |
|---|---|---|---|---|---|---|---|---|---|---|---|
| | MFSIM | 0.9534 | 0.9578 | 0.9772 | 0.9824 | 0.9817 | 0.9785 | 0.9802 | **0.9824** | 0.9814 | 0.9804 |
| 5 | MERGAS | 3.12 | 2.84 | 1.50 | 1.96 | 1.46 | 2.50 | 1.38 | 1.26 | **1.19** | 1.54 |
| | MSAM | 0.0862 | 0.0775 | 0.0427 | 0.0495 | 0.0395 | 0.0569 | 0.0373 | 0.0349 | **0.0329** | 0.0425 |
| | MFSIM | 0.8213 | 0.8676 | 0.9394 | 0.9661 | 0.9629 | 0.9655 | 0.9639 | 0.9614 | **0.9673** | 0.9648 |
| 25 | MERGAS | 14.96 | 9.50 | 4.55 | 2.91 | 3.31 | 2.94 | 2.87 | 3.08 | **2.50** | 2.77 |
| | MSAM | 0.3087 | 0.1753 | 0.1044 | 0.0684 | 0.0726 | 0.0671 | 0.0676 | 0.0709 | **0.0599** | 0.0668 |
| | MFSIM | 0.7174 | 0.7861 | 0.8974 | 0.9495 | 0.9439 | 0.9484 | 0.9464 | 0.9487 | **0.9542** | 0.9465 |
| 50 | MERGAS | 28.00 | 14.51 | 7.44 | 4.24 | 4.89 | 4.28 | 4.45 | 4.38 | **3.68** | 4.23 |
| | MSAM | 0.4785 | 0.2175 | 0.1438 | 0.0890 | 0.0925 | 0.0864 | 0.0944 | 0.0880 | **0.0768** | 0.0934 |
| | MFSIM | 0.6000 | 0.6821 | 0.8240 | 0.9188 | 0.9065 | 0.9209 | 0.9170 | 0.9100 | **0.9329** | 0.9163 |
| 100 | MERGAS | 48.42 | 20.83 | 11.98 | 6.54 | 7.58 | 6.66 | 6.52 | 7.01 | **5.51** | 6.52 |
| | MSAM | 0.6566 | 0.2700 | 0.1939 | 0.1183 | 0.1193 | 0.1147 | 0.1205 | 0.1297 | **0.0977** | 0.1265 |
| | MFSIM | 0.9338 | 0.9455 | 0.9690 | **0.9831** | 0.9774 | 0.9761 | 0.9787 | 0.9828 | 0.9782 | 0.9679 |
| [0-15] | MERGAS | 5.42 | 4.29 | 2.29 | 2.10 | 1.89 | 2.53 | 1.68 | **1.36** | 1.48 | 2.34 |
| | MSAM | 0.1358 | 0.1052 | 0.0610 | 0.0509 | 0.0487 | 0.0582 | 0.0438 | **0.0368** | 0.0395 | 0.0624 |
| | MFSIM | 0.8196 | 0.8642 | 0.9261 | **0.9766** | 0.9554 | 0.9523 | 0.9603 | 0.9714 | 0.9748 | 0.9507 |
| [0-55] | MERGAS | 18.46 | 10.41 | 5.56 | 2.37 | 3.86 | 4.19 | 3.22 | 2.37 | **2.05** | 4.73 |
| | MSAM | 0.3563 | 0.1879 | 0.1171 | 0.0572 | 0.0798 | 0.0961 | 0.0731 | 0.0581 | **0.0518** | 0.1226 |
| | MFSIM | 0.7471 | 0.8057 | 0.8837 | **0.9725** | 0.9377 | 0.9339 | 0.9473 | 0.9613 | 0.9689 | 0.9370 |
| [0-95] | MERGAS | 29.42 | 14.25 | 8.15 | 2.68 | 5.36 | 6.14 | 4.60 | 3.07 | **2.50** | 6.95 |
| | MSAM | 0.4899 | 0.2274 | 0.1466 | 0.0632 | 0.0973 | 0.1262 | 0.0962 | 0.0719 | **0.0604** | 0.1520 |
| | MFSIM | 0.9028 | 0.9229 | 0.9519 | 0.9783 | 0.9713 | 0.9693 | 0.9721 | **0.9790** | 0.9768 | 0.9744 |
| Corr. | MERGAS | 8.25 | 5.91 | 4.07 | 2.29 | 2.44 | 2.67 | 2.10 | 1.92 | **1.65** | 1.97 |
| | MSAM | 0.2049 | 0.1368 | 0.1106 | 0.0559 | 0.0593 | 0.0661 | 0.0540 | 0.0481 | **0.0436** | 0.0527 |
| | MFSIM | 0.8177 | 0.8621 | 0.9365 | **0.9663** | 0.9604 | 0.9649 | 0.9639 | 0.9619 | 0.9651 | 0.9582 |
| Strip. | MERGAS | 15.38 | 10.20 | 4.84 | 3.00 | 3.55 | 3.09 | 2.99 | 3.02 | **2.62** | 3.26 |
| | MSAM | 0.3152 | 0.1886 | 0.1101 | 0.0698 | 0.0794 | 0.0794 | 0.0705 | 0.0702 | **0.0623** | 0.0795 |

Table 9: Additional metrics on DCMall

**Ablation studies.** In this paragraph, we present different ablation studies, demonstrating in Table 10 that our two-layer model outperforms single-layer models. We also demonstrate the usefulness of our variant with weights $\beta_j$ in Table 11 when the noise variance varies a lot between different bands.

| Metrics | Noisy | Spec | SpecSpat | Spec + SpecSpat |
|---------|-------|------|----------|-----------------|
| MPSNR | 16.03 | 30.96 | 40.13 | **42.17** |
| MSSIM | 0.0502 | 0.6884 | 0.9533 | **0.9677** |
| MFSIM | 0.8100 | 0.9708 | 0.9849 | **0.9925** |
| MERGAS | 51.48 | 8.84 | 3.00 | **2.39** |
| MSAM | 0.7546 | 0.1300 | 0.1021 | **0.0547** |

Table 10: Combination of sparse coding layers: we denote by *Spec* the Spectral Sparse Coding layer and by *SpecSpat* the Spectral-Spatial Sparse Coding layer. This experiment was run on ICVL with $\sigma = 50$.

Table 11: Our model without/with band-wise noise estimator (NE) on ICVL with band-dependent Gaussian noise and stripes noise

|  | Metrics | T3SC | T3SC + NE |
|--|---------|------|-----------|
| [0-15] | MPSNR | 52.85 | **53.31** |
|  | MSSIM | 0.9963 | **0.9967** |
| [0-55] | MPSNR | 47.39 | **48.64** |
|  | MSSIM | 0.9890 | **0.9911** |
| [0-95] | MPSNR | 44.92 | **46.30** |
|  | MSSIM | 0.9821 | **0.9859** |
| Strip. | MPSNR | 44.68 | **44.74** |
|  | MSSIM | 0.9801 | **0.9805** |

## C   Visual Examples

Finally, we show additional visual examples in Figure 1 and 2.

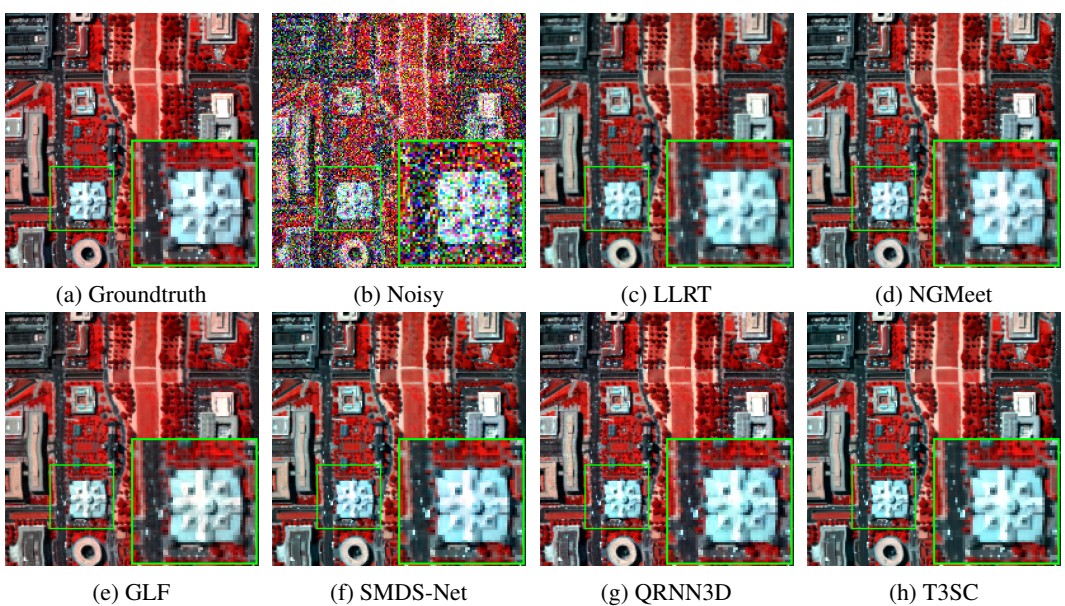

(a) Groundtruth    (b) Noisy    (c) LLRT    (d) NGMeet

(e) GLF    (f) SMDS-Net    (g) QRNN3D    (h) T3SC

Figure 1: Simulated Gaussian noise ($\sigma = 100$) on DCMall

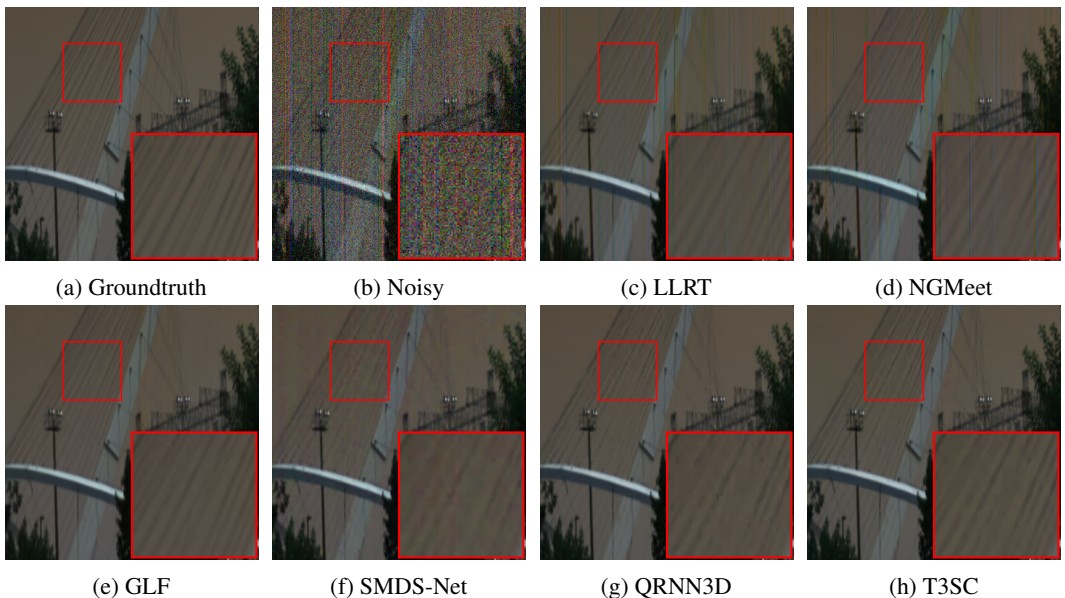

| (a) Groundtruth | (b) Noisy | (c) LLRT | (d) NGMeet |
| (e) GLF | (f) SMDS-Net | (g) QRNN3D | (h) T3SC |

Figure 2: Visual results for the denoising experiment with stripes noise on ICVL with bands 9, 15, 28.

## D  GPU ressources

The total number of GPU hours involved in this project is around 19k hours on NVIDIA Tesla V100 16Go, including preliminary experiments, model design, final experiments and running baseline methods.

## References

[1]  K. He, X. Zhang, S. Ren, and J. Sun. Delving deep into rectifiers: Surpassing human-level performance on imagenet classification. In *Proceedings of the IEEE International Conference on Computer Vision (ICCV)*, December 2015.