# OpenReview forum: "A Trainable Spectral-Spatial Sparse Coding Model for Hyperspectral Image Restoration"
_NeurIPS.cc/2021/Conference — NeurIPS 2021 Poster_

### Official Review · Reviewer_XbQS · 2021-07-06

**Rating:** 6
**Confidence:** 5

**Summary:**

This paper proposed a trainable spectral-spatial sparse coding model for hyperspectral image restoration. This paper proposed a hybrid approach based on sparse coding principles that retain the interpretability of classical techniques encoding domain knowledge with handcrafted image priors, while allowing to train model parameters end-to-end without massive amounts of data. The proposed model is claimed to be computationally efficient, and produce state-of-the-art restoration results.

**Limitations And Societal Impact:**

Please refer the main review

**Main Review:**

Trainable sparse coding has been utilized in nature image processing. But as far I know, this should be the first paper to introduce trainable sparse coding to hyperspectral image processing.

As an application work of trainable sparse coding for the hyperspectral image restoration, I have the following comments, regarding the review of the past work, the novelty of the proposed method, and the experimental results.

1. Regarding hyperspectral image restoration, low-rank matrix/tensor methods are the representative classical methods and achieved state-of-the-art performance. However, most of the related works are missing.

2. Very recently, deep learning-related methods have been widely used for hyperspectral image restoration. But what are the main challenges of deep learning for hyperspectral restoration? What is the progress of the related works? The authors summarized that insufficient training dataset is one challenge of the related topic. But many works have contributed to this, which are disappeared in the introduction of the related work.

3. The introduction of the proposed method is unclear. Why the proposed trainable sparse coding is suitable for hyperspectral restoration? The insight analysis is insufficient. Furthermore, it is hard to understand the proposed method from figure 1, and realize the proposed method from figure 1. I can understand, the trainable sparse coding can provide interpretability. But what is new for hyperspectral processing society?

4. It is better to choose the latest tensor-based methods for comparison to demonstrate the advantage of the proposed method. The illustrated results are also unreliable. As reported in the reference paper, QRNN3D can achieve 40.23dB for the case ICVL dataset with noise 50. Furthermore, the tensor-related methods can easily achieve 35dB for the case of Washington DC Mall with noise intensity 50. However, the reported results are lower than those from the reference papers.



**Time Spent Reviewing:**

4

---

> ### Author Response · Authors · 2021-08-10
> **Reply to reviewer XbQS**
>
> We thank the reviewer for their insightful feedback.
> * Regarding references, we agree that we have focused in large parts on deep learning and sparse coding models. We would be very happy to discuss additional
>     ones, but it would be helpful to provide precise references since we are unsure
>     what these related works are.
> * "But many works have contributed to this'" : We would be very grateful to discuss additional work, if precise references are given.
>      \item including tensor-based methods in the comparison. We have chosen to include
>      GLF, QRNN3D, which achieve SOTA results for learning-free and learning-based
>      approaches (and which include comparisons with tensor-based methods). Nevertheless, if a precise reference is given, we would be happy to include it.
> * " QRNN3D can achieve 40.23dB for the case ICVL dataset with noise 50".
>     Thank you for pointing this out: we have investigated the issue and now realize that there is an issue with our
>     training protocol (see comment for all reviewers).
> * Reference numbers for DCMall: these numbers were obtained by running the code of the reference methods and we believe them to be accurate. GLF reports higher numbers (greater than 35dB), but they do not compare the same object, since they apply a spectral projection on the clean image, reducing their rank. If a precise reference
>     for a tensor-based method is provided, we will be happy to investigate this question.

---

> > ### Comment · Reviewer_XbQS · 2021-08-20
> > **Additional comments**
> >
> > From the authors' reply, I have the following additional comments.
> >
> > 1. Regarding the performance of QRNN3D on ICVL dataset, in the revised results, QRNN3D achieved better results in the case of noise 50 and 100, meanwhile worse results in the case of noise 5 and 25. The explanation of why the proposed method can achieve better results in the low-noise case is insufficient. From the reviewer's experience, ICVL is an "easy" dataset, that's to say, a simple method (like Unet) can achieve enough good denoised results. Furthermore, ICVL dataset contains hundreds of images. In most cases, the authors of different paper always choose different training and test dateset, resulting in the different reported results. From this side, ICVL dataest is not suitable as the baseline to testify the performance of different methods.
> >
> > 2. Regarding tensor-related methods, KBR(TPAMI2018), LLRT(CVPR2017), and NGmeet(2020TPAMI) stand for the (nearly) SOTA  performance for HSI denoising. Furthermore, these papers regard CAVE dataset as the baseline dataset. If the authors want to claim that the proposed method achieved SOTA, the proposed method should beat these methods on CAVE dataset.
> >
> > 3. Regarding the performance of GLF, I don't understand what is the meaning of "do not compare the same object". Generally, the main contribution of GLF is the spectral projection to explore the global spectral low-rank property. What's more, the rank of spectral dimension should be adjusted. The parameters of the compared methods also need to be taken into consideration.
> >
> > As pointed out by the other reviewers, the proposed method is a  simple two-stage strategy from a new perspective. It has much room for improvement, and is not expected to achieve SOTA denoising performance. However, if the authors claimed SOTA, the experiments should be more convincing, i.e., the proposed method needs to be compared to the previous SOTA on the standard dataset.

---

> > > ### Author Response · Authors · 2021-08-22
> > > **Response**
> > >
> > > Thank you very much for engaging the discussion and for the additional feedback, which is very useful. This response addresses points 1 and 3 and only partly point 2 (we are just starting to investigate the performance on our method on the CAVE dataset, and will post an update as soon as possible).
> > >
> > > We first discuss technical points, before discussing the definition of SOTA for HSI denoising (a discussion, which we believe should be included in the paper). We would be happy to provide additional explanations if needed.
> > >
> > > **QRNN3D vs Ours on ICVL**: We agree that there was no reason for our method to be better for low-noise regimes and worse for high-noise regimes.  After running both approaches with the updated protocol (see post "Final comparison with QNN3D on ICVL"), the results are much more consistent, and we are confident in claiming that our approach does significantly better than QRNN3D in both noise regimes. We thank the reviewer for pointing out the initial inconsistency, since it has allowed to significantly improve our results.
> > >
> > > **Regarding the ICVL dataset**: We would like to provide the following clarifications:
> > > - our original plan was to use the same train/test split of QRNN3D, but we realized
> > >   that their split was problematic because several test images were almost identical
> > >   to each other (with only a small object chaning in the center of the scene,
> > >   see for instance the sequence starting at objects_0924-1550 from the ICVL
> > >   website: http://icvl.cs.bgu.ac.il/hyperspectral/). Therefore, we decided to
> > >   remove these images and generate a better split with enough diversity within
> > >   train and test (and no leakage of information between train/test).
> > >   We agree that this is unfortunate that the HSI literature did not converge
> > >   to a standard train/test split for ICVL, but we hope that our effort will be
> > >   helpful to achieve this goal.
> > > - we respectfully disagree that ICVL is a bad baseline: it has indeed some issues
> > > such as the one mentioned above, but it is one of the very few datasets with
> > > a significant number of clean images. It is thus true that the dataset is *well adapted to learning-based approaches*, which does not mean it is a bad
> > > dataset (see then discussion below on "Claiming SOTA"). We also would like to
> > > underline that our approach does significantly better than simple U-nets. As
> > > shown above, we do better than QRNN3D with an apple-to-apple comparison, and we
> > > refer to Table III of https://arxiv.org/pdf/2003.04547.pdf regarding the
> > > comparison of QRNN3D to other deep-learning models such as HSID-CNN (+4dB for
> > > QRNN3D on $\sigma=50$) and MemNet (+0.5dB).
> > >
> > > **Running tensor-based methods on ICVL**: Following the reviewer's advice, we
> > > have conducted experiments on ICVL with the method NGMeet (we will also post an
> > > update when we receive the results for LRRT), with the code available here
> > > https://github.com/quanmingyao/NGMeet.  The results for uniform $\sigma$ are
> > > very good (better than GLF and even better than us for $\sigma=5$) for this
> > > method and we agree that they should be included in the paper.
> > >
> > > | sigma | NgMeet | GLF | QRNN3D (2b) | Ours (2b) |
> > > |---|---|---|---|---|
> > > | 5  |  __52.75__  | 51.25 | 50.52  | 52.26 |
> > > | 25 | 44.58 |  43.16 | 43.53  |  __44.9__ |
> > > | 50 | 40.88 | 39.26 | 41.34 | __41.69__ |
> > > | 100 | 37.37  | 34.79|  36.82 | __38.39__ |
> > >
> > > We also conducted experiments with NgMeet/LRRT for other types of noise, but
> > > the implementation mentioned above was not adapted for such settings and we
> > > obtained poor results (not reported here).
> > >
> > > **Regarding the CAVE dataset**: We agree that this dataset is interesting. We will
> > > follow the reviewer's suggestion and do our best to report results on this dataset
> > > as soon as possible, before the discussion period ends.
> > >
> > > **Performance of GLF on DC Mall**: We apologize if our original message was not
> > > clear enough. We claim that the numbers reported on DC Mall in the GLF paper
> > > cannot be compared with the numbers of other papers, unless a specific
> > > pre-processing is applied *on the ground truth image*.  The corresponding
> > > sentence from the GLF paper provides the right explanation.
> > >
> > > > Furthermore, the remaining spectral vectors were then projected onto the
> > > signal subspace of dimension 8 learned via performing SVD on correlation
> > > matrix. The obtained HSI is considered the clean HSI to which Gaussian i.i.d. noise ... is added.
> > >
> > > In other workds, GLF changes the ground-trugh in their evaluation protocol.
> > >
> > > **Performance of NGMeet/LRRT on DC Mall**: We thank the reviewer for pointing
> > > out the NGMeet paper. Indeed, NGMeet reports about 35dB on this dataset for
> > > $\sigma=50$. After investigation, we noticed that the test image (subset of the
> > > original large Washington DC Mall dataset) used in (KBR, LRRT, NGMeet), was
> > > *different* than the one used in (HSI-CNN, 3D-ADNet), which we use in our
> > > paper. This is unfortunately yet another illustration of issues from the HSI
> > > denoising literature, which has not yet adopted standard benchmarks.
> > > These two test images are very different from each other, making the comparison
> > > difficult. We are currently running the code from NGMeet on our test image and
> > > we currently obtain $31.63$dB for $\sigma=50$ and $30.7$dB for LRRT, which
> > > suggests that our test image may be more difficult to process (running this
> > > code on the other test image gives indeed a PSNR close to 35dB).  Another
> > > explanation for such a difference may be due to the normalization step used by
> > > HSI-CNN, 3D-ADNet and ours, before adding noise: we perform a normalization
> > > *per band* to obtain values in [0,255], whereas NGMeet performs a normalization
> > > *per image*.
> > >
> > > **On claiming SOTA**: The previous discussion raises some questions about
> > > the definition of SOTA for HSI denoising. We believe that it is legitimate to
> > > consider separately learning-free from learning-based approaches.  These two
> > > classes of approaches have very different requirements/characteristics, making
> > > one class more relevant than the other one in some scenarios, and less in others.
> > > The table below summarizes their characteristics, displaying advantages and drawbacks
> > > of both approaches.
> > >
> > > |                  | training data  | training time | inference time | deployment to new datasets |
> > > |---|---|---|---|---|
> > > | learning-free   | no requirement | no training   |  slow          | easy |
> > > | learning-based | need data      | slow          |  fast          | more complicated |

---

### Official Review · Reviewer_ujx6 · 2021-07-08

**Rating:** 7
**Confidence:** 4

**Summary:**

The authors propose a denoising technique for hyperspectral images that is a combination of learning and model-based techniques. In particular, a dictionary learning technique using low-rank and sparsity priors is developed.

**Limitations And Societal Impact:**

The authors addressed the impact of their work.

**Main Review:**

Overall, the paper is well-written and provides all the necessary backgroud to follow the explanation of the method. The problem of denoising hyperspectral images is important and often neglected by the community due to the challenges in finding large datasets. The necessity to carefully exploit spatial and spectral correlation and specific kinds of noise also make it significantly different from traditional RGB image denoising. The authors address all these challenges leveraging components which are in part well-known (e.g. the use of LISTA) but overall in an innovative manner. Addressing generalization of a single model to multiple sensors is also interesting in itself.

Experiments cover a wide range of noise models and state-of-the-art methods and are adequate at characterizing the performance of the proposed method. However, I would have liked to see additional metrics besides SNR and SSIM, in particular regarding spectral quality (ERGAS, SAM, ...).

**Time Spent Reviewing:**

1

---

> ### Author Response · Authors · 2021-08-10
> **Reply to reviewer ujx6**
>
> We thank the reviewer for their encouraging comments.
> Additional metrics can be found in Table 8 of the supplementary material for ICVL (MFSIM, MERGAS, MSAM) for each noise pattern. We will clarify this in the main paper, if it is accepted.

---

### Official Review · Reviewer_xNh4 · 2021-07-17

**Rating:** 6
**Confidence:** 3

**Summary:**

The paper proposes a deep learning based algorithm for hyperspectral image restoration that is interpretable, in the sense that the learning process separates the structure of the spatial and spectral dimensions. This allows the method to leverage the significant amount of available data for the spatial dimensions (e.g, image denoising). The method is computationally efficient.

**Limitations And Societal Impact:**

A single limitation is listed. The impact describes applications desired and undesired by the authors.

**Main Review:**

There are many efforts in ML applied to imaging that have separated the learning of the spatial and spectral structure, as listed in "Benchmarked models"; however, the comparison with these baselines is fairly limited to one experiment in the supplemental material.

It is not sufficiently clear what the interpretability of the proposed method entails. How is the result of the restoration interpreted to obtain additional information? The fact that different dimensions are identified to have different known structures may not fit the definition of interpretable ML, which I understand to be that the algorithm provides knowledge in the structure it identifies; it seems to me here that the structure is already enforced in the learning model.

The approach of Section 3.4 is designed for nonuniform noise, but there is nothing intrinsic in it regarding hyperspectral imaging.

The computational efficiency does not account for training time. This would give a sense of how the training complexity compares to the computational complexity of the other algorithms and at what scaling point the algorithm becomes efficient across the board.

Minor typos: Table 1 caption "first two" -> "first four".

**Time Spent Reviewing:**

4 hours

---

> ### Author Response · Authors · 2021-08-10
> **Reply to reviewer xNh4**
>
> We thank the reviewer for their insightful feedback.
> * We could include the results for 3D-ADNet and HSID-CNN in Table 5 of the supplementary material, by reporting the numbers from their papers as they use the same protocol and train/test split on DCMall. We could not include these approaches in the ICVL benchmark because of the computational cost which was too large for our budget: each deep learning method evaluated requires training on 9 different noise patterns, multiplied by 5 random seeds (without taking into account hyper-parameter tuning). As such, we decided to keep to the most relevant ones (QRNN3D which achieves impressive SOTA results and SMDS-Net).
> * Regarding interpretability, we are planning to investigate our model for spectral unmixing: the output code of the first layer represents a decomposition of each pixel into elements of the spectral dictionary $\mathbf{D}_1$, which can be linked to the proportion of each endmember/material present in the pixel.    However, we agree that this requires significant additional work which is beyond the scope of our paper.
> *  We argue that nonuniform noise is relevant for hyperspectral imaging.
>     Indeed, hyperspectral noise can be modeled as the sum of "thermal noise"(signal-independent) and "photon noise" (signal-dependent), see _Mixed Poisson-Gaussian noise model based sparse denoising for hyperspectral imagery_, Ye \& Qian, 2012.
>     As a result, noise variances are varying throughout the hyperspectral bands, as mentioned in Section 1.5.2 of _Noise Reduction in Hyperspectral Imagery: Overview and Application_, Rasti \& al, 2018.
>     Band-dependent gaussian noise is an approximation of this phenomenon, (also used in SMDS-Net paper).
> * Discussing training time is indeed important. In our case, it is about 24h on a single GPU for ICVL. Note that this can be performed offline in general.
>
> If the paper is accepted, the above discussion will be included.

---

> > ### Comment · Reviewer_xNh4 · 2021-08-19
> > **Additional results**
> >
> > Thank you for your response. Can you provide here the comparison (with the result from their papers) that would be added to the manuscript? Can you also provide the quotes on computation times for the different approaches that would be added to the manuscript, including training and testing?
> >
> > The description of interpretability would then be better located within "Broader Impact", or a section of the paper describing further work?
> >
> > I agree that nonuniform noise is relevant to hyperspectral imaging; my point is that the approach provided addresses nonuniform noise in all cases where it appears, not only for hyperspectral imaging; furthermore it is not clear what role the "interpretability" mentioned in the title of the subsection plays here.

---

> > > ### Author Response · Authors · 2021-08-22
> > > **Response**
> > >
> > > Thank you for engaging the discussion and for providing additional feedback. We
> > > hope our response addresses your concerns and that you will reevaluate our work positively. We will be happy to provide more details or clarifications, if needed.
> > >
> > > **Additional Comparisons**
> > > Following the discussion with reviewer XbQS, a few tensor-based baselines will
> > > be added (at least for NGMeet and LRRT), since they amount of computation required
> > > for these approaches is reasonable. We refer to the on-going discussion with
> > > reviewer XbQS for more details and the numbers that will be added to the
> > > comparison.
> > > We are also planning to include the discussion on training protocols (see post above), along with the updated results.   Regarding the numbers reported from the 3D-ADNet and HSID-CNN
> > > papers, we are only able to include them for the Washington DC Mall dataset
> > > (see Table 5 in supplementary material), since they use the same evaluation
> > > protocol as ours.
> > >
> > > **Computational time**
> > > Regarding the computation times, we will make the following clarification in
> > > the caption describing inference time: ```Unlike GLF, NGMeet, and LRRT,
> > > learning-based approaches such as QRNN3D and our approach require a training
> > > procedure, which may conducted offline. Such a training step is relatively
> > > costly and was about 24 hours for both QRNN3D and our method, when using a
> > > single GPU (Titan RTX).'''  We will also include the inference times for NGMeet
> > > and LRRT (which have the same order of magnitude as GLF on CPU).
> > > We agree that such a discussion is important: deep learning-based approaches are typically much faster at inference time, but they require a training step which is indeed costly and may require parameter tuning (in particular the learning rate).
> > >
> > > **Interpretability**
> > > We thank the reviewer for the suggestion. We are planning to include the discussion
> > > above in a paragraph about Future work.
> > >
> > > **Non-uniform noise**
> > > We thank the reviewer for the clarification and we agree that the title of Section 3.4
> > > may be confusing. We will replace it by ``Adapting to Band-Dependent Noise''.

---

### Official Review · Reviewer_QMp1 · 2021-07-19

**Rating:** 6
**Confidence:** 4

**Summary:**

This paper proposes a denoising method for hyperspectral images in a supervised and unsupervised manner. The author proposed a trainable sparse coding model with two-steps dictionary learning. Their method can also process different HSI with different number of bands. The evaluation outperforms the baselines in the majority of scenes.

**Limitations And Societal Impact:**

lthough the technical contribution is limited, I am still surprised that this simple two-stage dictionary learning method can outperform state-of-arts. If the authors compare their methods with traditional dictionary learning denoising on 2D-profiles of the hyperspectral cube, it's worth accepted as an experimental paper.

**Main Review:**

1. Originality: Hyperspectral denoising is an interesting topic. It is well known that hyperspectral cube in nature is sparse and can be decomposed into a few basis. This is why we use dictionary learning for hyperspectral restoration and reconstruction. The related work is adequately cited. It's kind of a novel combination of LISTA algorithm, dictionary learning, and band-dependent noise analysis.
2. Quality: The majority of their claim is supported by experimental results. It is a complete work that covers different noise patterns, blind or non-blind denoising, a variety of public datasets, synthetic and real noise. For the experiment design, as the authors claim their key contribution is based on the extension of dictionary learning, they should compare with hyperspectral dictionary learning denoising [12]. Also, there could be another ablation study of applying dictionary learning along wavelength axis slices to support their band-dependent noise assumption in real data, and the necessity of their method.
3. Clarity: It is well written and organized. It has a sufficient description of details of experiments and comprehensive visual and quantitative results.
4. Significance: The results are important. If my concern raised in quality (2) can be clarified, it is a strong experimental evidence of superior performance among deep learning based methods. Others are likely to use and build based on this idea. The LISTA algorithm also facilitates the usage of GPU and speeds up the inference, as shown in Table 2. This is an advantage for large-scale deployment.

**Time Spent Reviewing:**

1hour

---

> ### Author Response · Authors · 2021-08-10
> **Reply to reviewer QMp1**
>
> We thank the reviewer for their useful feedback and encouraging comments.
>
> Regarding item 2, indeed our approach builds up on dictionary learning. We agree that comparing our approach to [12] is relevant. Therefore, please find below the results of [12] on ICVL dataset with different noise patterns.
>
> |   sigma  |Ours   | [12] (K-SVD)   |
> |---|---|---|
> | 5  |  __51.64__  | 49.45  |
> | 25  |  __43.77__ |  41.54 |
> | 50 |   __40.04__ | 38.04 |
> | 100 | __36.27__  |  34.46 |
> | [0-15]  | __50.17__  | 34.37  |
> | [0-55]  | __44.23__ | 23.90 |
> | [0-95] |  __42.67__ | 19.57 |
> | Corr. |  __48.66__ | 30.47 |
> | Strip. | __36.00__ | 24.02 |
>
>
>  We also agree that traditional dictionary learning methods applied band-wise is an interesting baseline. For that, we believe that the comparison with the state-of-the-art dictionary learning method [12] for hyperspectral imaging, suggested by the reviewer, illustrates the importance of a joint processing of multispectral bands, since this point has been well established in the dictionary learning literature for HSI.
>
> Note that updated results for our method are available in the post above with different training protocols, following a remark from Reviewer XbQS.

---

### Author Response · Authors · 2021-08-10
**Update of QRNN3D**

Following a comment from one reviewer, we have noticed an issue with our training protocol, reducing the performance of learning-based approaches.

  In our current protocol, models only see 80k different HSI patches whereas in QRNN3D reference paper, their model sees about 1.6M patches for training under known $\sigma$. With more training data, the performance of QRNN3D significantly improves, see table below (to be compared to Table 1 in our paper). Whereas our model (with the 80k-patches protocol) still performs better than QRNN3D (with 1.6M patches) for small sigma, it is now behind for large ones.

  We are going to investigate if our model also improves with additional training data and will post results here if we manage to obtain them soon. In the meantime, we believe it is important that the reviewers reassess the performance of our model in view of this updated numbers. We believe the current numbers to be interesting as they
  illustrate the ability of our method to achieve competitive performance with only a small training set.

|   sigma |  QRNN3D    |
|---|---|
| 5  |   50.48  |
| 25  |    43.47 |
| 50 |   41.25 |
| 100 |   36.48 |
| [0-15]  |  47.91  |
| [0-55]  | 45.81 |
| [0-95] |  43.06 |
| Corr. |   48.69|
| Strip. |  42.47 |

---

> ### Author Response · Authors · 2021-08-21
> **Final comparison with QNN3D on ICVL**
>
> As promised above, we have run again the experiments on ICVL, by using the training protocol of QRNN3D. We refer to this protocol as (2). The previous protocol is denoted by (1). This new protocol significantly improves the results because it generates a larger number of patches during training. As a consequence,
>  - there is no discrepancy anymore between the numbers reported in the original QRNN3D paper, and the ones we report here.
>  - both our method and QRNN3D improve by using (2) instead of (1), leaving the ranking between the two methods unchanged.
>
> We also added a variant of (2) that processes larger 256x256 or 512x512 image blocks instead of the original 56x56 blocks, which further improves the results for high-noise regimes. We call such a variant (2b).
>
> | sigma | QRNN3D (1) |Ours (1)| QRNN3D (2) | Ours (2) | QRNN3D (2b) | Ours (2b) |
> |---|---|---|---|---|---|---|
> | 5  |43.81 | 51.64  | 50.48  | __52.26__ | 50.52 | __52.26__
> | 25  |40.69 |  43.77 |  43.47 | 44.6 | 43.53| __44.9__
> | 50 | 38.95 | 40.04 | 41.25 | 41.07 | 41.34 | __41.69__
> | 100 | 36.19  |  36.27 | 36.48 | 37.34 | 36.82 | __38.39__
> | [0-15]  | 44.46 | 50.17  | 47.91  | __52.62__ | 47.93 | 52.28
> | [0-55]  | 40.23 | 44.23 | 45.81 |  47.3 | 45.78 | __47.67__
> | [0-95] |  38.19  | 42.67 | 43.06 | 44.49 | 43.02 | __45.4__
> | Corr. |   41.78 | 48.66 | 48.69|  49.59 | 48.7 | __49.66__
> | Strip. | 33.87 | 36.00 | 42.47 | 43.17 | 42.27 | __43.75__
>
> We thank again the reviewers for pointing out the original discrepancy, which has allowed us to significantly improve our results!
>
> (Note that this post has been edited to add the results with the protocol "(2b)")

---

> > ### Author Response · Authors · 2021-08-25
> > **Comparison with NGMeet on CAVE**
> >
> > As promised to reviewer XbQS, we have conducted an experiment on CAVE, and made
> > a comparison with NGMeet.  For other baselines, we refer to the following
> > survey https://arxiv.org/pdf/2011.03462.pdf who also used the 32 512x512x31 images
> > from the CAVE dataset for the evaluation.
> >
> > Since evaluation on all images is problematic for supervised learning, we
> > conducted the following experiment, where we split the dataset into four folds
> > of 8 images. Then, we learn four models, such that each model is trained on 3
> > folds (24 images) and is evaluated on the remaining fold (8 images). This
> > allows us to evaluate the denoising performance (in terms of MPSNR) for each
> > image. We also used the same learning rate as in the ICVL experiment and did
> > not perform any hyper-parameter tuning here (which is a good sanity check that
> > the choice of hyper-parameters is relatively robust across datasets).
> >
> > The MPSNR results are given in the table below for uniform noise (as for ICVL,
> > we did not obtain good results for NGMeet for non-uniform noise since this
> > method was not designed for such a setting).
> >
> > |sigma| NGMeet | Ours |
> > |---|---|---|
> > |5  | 47.96 | __49.16__ |
> > |25 | 42.44 | __42.77__ |
> > |50 | 38.89 | __39.7__ |
> > |100 | 34.99 | __36.48__ |
> >
> > We are relatively confident with these results since
> >  - they are consistent with those reported in https://arxiv.org/pdf/2011.03462.pdf
> >  - from a personal communication with one author of the NGMeet paper, we have the confirmation that 38.89 is correct for $\sigma=50$ when performing the evaluation on the whole images.
> >
> > The conclusions from this experiment are that NGMeet indeed performs well one
> > this dataset, but that a supervised learning approach like ours may bring some
> > benefits in terms of denoising performance and speed (and it can also handle
> > gracefully complex noise patterns).  Of course, supervised learning comes at
> > some cost, notably the need to obtain training data generated with the same
> > sensor (see discussion in discussion to reviewer XbQS), which should of
> > course be mentioned here.
> >
> > We thank the reviewers for all the interactions, and for having raised their
> > scores.

---

### Decision · Program_Chairs · 2021-09-27

**Decision:**

Accept (Poster)

**Comment:**

This work presents a combined sparse-coding and deep learning approach to achieve practical and efficient de-noising of hyperspectral images.  Following a discussion with the reviewers, the authors were able to clarify a number of key points relating to compute time and the relationship to competing methods. More importantly, the authors fixed an error identified by one reviewer and have indicated new results that will be included instead of those currently in the paper. With these corrections, the reviewers have updated their scores and I therefore recommend this work be accepted to NeurIPS.